# Provably Doubly Accelerated Federated Learning: The First Theoretically Successful Combination of Local Training and Communication Compression

## Abstract

In federated learning, a large number of users collaborate to learn a global model. They alternate local computations and two-way communication with a distant server. Communication, which can be slow and costly, is the main bottleneck in this setting. To reduce the communication load and therefore accelerate distributed gradient descent, two strategies are popular: 1) communicate less frequently; that is, perform several iterations of *local* computations between the communication rounds; and 2) communicate *compressed* information instead of full-dimensional vectors. We propose the first algorithm for distributed optimization and federated learning, which harnesses these two strategies jointly and converges linearly to an exact solution in the strongly convex setting, with a doubly accelerated rate: our algorithm benefits from the two acceleration mechanisms provided by local training and compression, namely a better dependency on the condition number of the functions and on the dimension of the model, respectively.

## 1 Introduction

Federated Learning (FL) is a novel paradigm for training supervised machine learning models. Initiated a few years ago (Konečný et al., 2016a;b; McMahan et al., 2017; Bonawitz et al., 2017), it has become a rapidly growing interdisciplinary field. The key idea is to exploit the wealth of information stored on edge devices, such as mobile phones, sensors and hospital workstations, to train global models, in a collaborative way, while handling a multitude of challenges, like data privacy (Kairouz et al., 2021; Li et al., 2020a; Wang et al., 2021). In contrast to centralized learning in a datacenter, in FL, the parallel computing units have private data stored on each of them and communicate with a distant orchestrating server, which aggregates the information and synchronizes the computations, so that the process reaches a consensus and converges to a globally optimal model. In this framework, communication between the parallel workers and the server, which can take place over the internet or cell phone network, can be slow, costly, and unreliable. Thus, communication dominates the overall duration and cost of the process and is the main bottleneck to be addressed by the community, before FL can be widely adopted and applied in our daily lives.

The baseline algorithm of distributed Gradient Descent (GD) alternates between two steps: one round of parallel computation of the local function gradients at the current model estimate, and one round of communication of these gradient vectors to the server, which averages them to form the new estimate for the next iteration. To decrease the communication load, two strategies can be used: 1) communicate less frequently, or equivalently do more *local computations* between successive communication rounds; or 2) *compress* the communicated vectors. We detail these two strategies in Section 1.3. In this paper, we combine them, within a unified framework for randomized communication, and derive a new algorithm named CompressedScaffnew, with local training and communication compression. It is variance-reduced (Hanzely & Richtárik, 2019; Gorbunov et al., 2020a; Gower et al., 2020), so that it converges to an exact solution, and provably benefits from the two mechanisms: the convergence rate is doubly accelerated, with a better dependency on the condition number of the functions and on the dimension of the model, in comparison with GD. In the

remainder of this section, we formulate the convex optimization problem to solve, we propose a new model to characterize the communication complexity, and we present the state of the art.

## 1.1 FORMALISM

We consider a distributed client-server setting, in which $n \geq 2$ clients perform computations in parallel and communicate back and forth with a server. We study the convex optimization problem:

$$\underset{x \in \mathbb{R}^d}{\text{minimize}} \ f(x) := \frac{1}{n} \sum_{i=1}^{n} f_i(x), \tag{1}$$

where each function $f_i : \mathbb{R}^d \to \mathbb{R}$ models the individual cost of client $i \in [n] := \{1, \ldots, n\}$, based on its underlying private data. The number $n$ of clients, as well as the dimension $d \geq 1$ of the model, are typically large. This problem is of key importance as it is an abstraction of empirical risk minimization, the dominant framework in supervised machine learning.

For every $i \in [n]$, the function $f_i$ is supposed $L$-smooth and $\mu$-strongly convex,[1] for some $L \geq \mu > 0$ (a sublinear convergence result is derived in the Appendix for the merely convex case, i.e. $\mu = 0$). Thus, the sought solution $x^\star$ of equation 1 exists and is unique. We define $\kappa := \frac{L}{\mu}$. We focus on the strongly convex case, because the analysis of linear convergence rates in this setting gives clear insights and allows us to deepen our theoretical understanding of the algorithmic mechanisms under study; in our case, local training and communication compression. The analysis of algorithms converging to a stationary point with nonconvex functions relies on significantly different proof techniques (Karimireddy et al., 2021; Das et al., 2022), so the nonconvex setting is out of the scope of this paper.

To solve equation 1, the baseline algorithm of Gradient Descent (GD) consists in the simple iteration $x^{t+1} := x^t - \frac{\gamma}{n} \sum_{i=1}^{n} \nabla f_i(x^t)$, for some stepsize $\gamma \in (0, \frac{2}{L})$. That is, at iteration $t \geq 0$, $x^t$ is first broadcast by the server to all clients, which compute the gradients $\nabla f_i(x^t) \in \mathbb{R}^d$ in parallel. These vectors are then sent by the clients to the server, which averages them and updates the model estimate. It is well known that for $\gamma = \Theta(\frac{1}{L})$, GD converges linearly, with iteration complexity $\mathcal{O}(\kappa \log \epsilon^{-1})$ to reach $\epsilon$-accuracy. Since $d$-dimensional vectors are communicated at every iteration, the communication complexity of GD in number of reals is $\mathcal{O}(d\kappa \log \epsilon^{-1})$. Our goal is a twofold acceleration of GD, with a better dependency to both $\kappa$ and $d$ in this communication complexity. We want to achieve this goal by leveraging the best of the two popular mechanisms of local training and communication compression.

## 1.2 ASYMMETRIC COMMUNICATION REGIME

**Uplink and downlink communication**. We call *uplink communication* (UpCom) the parallel transmission of data from the clients to the server and *downlink communication* (DownCom) the broadcast of the same message from the server to all clients. UpCom is usually significantly slower than DownCom, just like uploading is slower than downloading on the internet or cell phone network. This can be due to the asymmetry of the service provider's systems or protocols used on the communication network, or cache memory and aggregation speed constraints of the server, which has to decode and average the large number $n$ of vectors received at the same time during UpCom.

**Communication complexity**. We measure the UpCom or DownCom complexity as the expected number of communication rounds needed to estimate a solution with $\epsilon$-accuracy, *multiplied by* the number of real values sent during a communication round between the server and any client. Thus, the UpCom or DownCom complexity of GD is $\mathcal{O}(d\kappa \log \epsilon^{-1})$. We leave if for future work to refine this model of counting real numbers, to take into account how sequences of real numbers are quantized into bitstreams, achieving further compression (Horváth et al., 2022; Albasyoni et al., 2020).

**A model for the overall communication complexity**. Since UpCom is usually slower than Down-Com, we propose to measure the *total communication* (TotalCom) complexity as a weighted sum of

---

[1] A function $f : \mathbb{R}^d \to \mathbb{R}$ is said to be $L$-smooth if it is differentiable and its gradient is Lipschitz continuous with constant $L$; that is, for every $x \in \mathbb{R}^d$ and $y \in \mathbb{R}^d$, $\|\nabla f(x) - \nabla f(y)\| \leq L\|x - y\|$, where, here and throughout the paper, the norm is the Euclidean norm. $f$ is said to be $\mu$-strongly convex if $f - \frac{\mu}{2}\|\cdot\|^2$ is convex. We refer to Bauschke & Combettes (2017) for such standard notions of convex analysis.

the two UpCom and DownCom complexities: we assume that the UpCom cost is 1 (unit of time per transmitted real number), whereas the downCom cost is $c \in [0, 1]$. Therefore,

$$\text{TotalCom} = \text{UpCom} + c.\text{DownCom}. \tag{2}$$

A symmetric but unrealistic communication regime corresponds to $c = 1$, whereas ignoring downCom and focusing on UpCom, which is usually the limiting factor, corresponds to $c = 0$. We will provide explicit expressions of the parameters of our algorithm to minimize the TotalCom complexity for any given $c \in [0, 1]$, keeping in mind that realistic settings correspond to small values of $c$. Thus, our model of communication complexity is richer than only considering $c = 0$, as is usually the case.

## 1.3 COMMUNICATION EFFICIENCY IN FL: STATE OF THE ART

Two approaches come naturally to mind to decrease the communication load: *Local Training* (LT), which consists in communicating less frequently than at every iteration, and *Communication Compression* (CC), which consists in sending less than $d$ floats during every communication round. In this section, we review existing work related to these two strategies.

### 1.3.1 LOCAL TRAINING (LT)

LT is a conceptually simple and surprisingly powerful communication-acceleration technique. It consists in the clients performing multiple local GD steps instead of only one, between successive communication rounds. This intuitively results in "better" information being communicated, so that less communication rounds are needed to reach a given accuracy. As shown by ample empirical evidence, LT is very efficient in practice. It was popularized by the FedAvg algorithm of McMahan et al. (2017), in which LT is a core component. However, LT was heuristic and no theory was provided in their paper. LT was analyzed in several works, in the homogeneous, or i.i.d. data, regime (Haddadpour & Mahdavi, 2019), and in the heterogeneous regime, which is more representative in FL (Khaled et al., 2019; 2020; Stich, 2019; Woodworth et al., 2020; Li et al., 2020b; Gorbunov et al., 2021; Glasgow et al., 2022). It stands out that LT suffers from so-called client drift, which is the fact that the local model obtained by client $i$ after several local GD steps approaches the minimizer of its local cost function $f_i$. The discrepancy between the exact solution $x^\star$ of equation 1 and the approximate solution obtained at convergence of LT was characterized in Malinovsky et al. (2020). This deficiency of LT was corrected in the Scaffold algorithm of Karimireddy et al. (2020) by introducing control variates, which correct for the client drift, so that the algorithm converges linearly to the exact solution. S-Local-GD (Gorbunov et al., 2021) and FedLin (Mitra et al., 2021) were later proposed, with similar convergence properties. Yet, despite the empirical superiority of these recent algorithms relying on LT, their communication complexity remains the same as vanilla GD, i.e. $\mathcal{O}(d\kappa \log \epsilon^{-1})$.

It is only very recently that Scaffnew was proposed by Mishchenko et al. (2022), a LT algorithm finally achieving $\mathcal{O}(d\sqrt{\kappa} \log \epsilon^{-1})$ accelerated communication complexity. In Scaffnew, communication is triggered randomly with a small probability $p$ at every iteration. Thus, the expected number of local GD steps between two communication rounds is $1/p$. By choosing $p = 1/\sqrt{\kappa}$, the optimal dependency on $\sqrt{\kappa}$ instead of $\kappa$ is obtained. Thus, Scaffnew is an important milestone, as it provides the theoretical confirmation that LT is a communication acceleration mechanism. In this paper, we propose to go even further and tackle the multiplicative factor $d$ in the complexity of Scaffnew.

### 1.3.2 COMMUNICATION COMPRESSION (CC)

To decrease the communication complexity, a widely used strategy is to make use of (lossy) compression; that is, a possibly randomized mapping $\mathcal{C} : \mathbb{R}^d \to \mathbb{R}^d$ is applied to the vector $x$ that needs to be communicated, with the property that it is much faster to transfer $\mathcal{C}(x)$ than the full $d$-dimensional vector $x$. A popular sparsifying compressor is rand-$k$, for some $k \in [d] := \{1, \ldots, d\}$, which multiplies $k$ elements of $x$, chosen uniformly at random, by $d/k$, and sets the other ones to zero. If the receiver knows which coordinates have been selected, e.g. by running the same pseudo-random generator, only these $k$ elements of $x$ are actually communicated, so that the communication complexity is divided by the compression factor $d/k$. Another sparsifying compressor is top-$k$, which keeps the $k$ elements of $x$ with largest absolute values unchanged and sets the other ones to zero. Some compressors, like rand-$k$, are unbiased; that is, $\mathbb{E}[\mathcal{C}(x)] = x$ for every $x \in \mathbb{R}^d$, where $\mathbb{E}[\cdot]$

denotes the expectation. On the other hand, compressors like top-$k$ are biased (Beznosikov et al., 2020).

The variance-reduced algorithm DIANA (Mishchenko et al., 2019), later extended in several ways (Horváth et al., 2022; Gorbunov et al., 2020a; Condat & Richtárik, 2022), is a major contribution, as it converges linearly with a large class of unbiased compressors. For instance, when the clients use independent rand-1 compressors for UpCom, the UpCom complexity of DIANA is $\mathcal{O}\big((\kappa(1 + \frac{d}{n}) + d)\log \epsilon^{-1}\big)$. If $n$ is large, this is much better than with GD. Algorithms converging linearly with biased compressors have been proposed recently, like EF21 (Richtárik et al., 2021; Fatkhullin et al., 2021; Condat et al., 2022), but the theory is less mature and the acceleration potential not as clear as with unbiased compressors. We summarize existing results in Table 1. Our algorithm CompressedScaffnew benefits from CC with specific unbiased compressors, with even more acceleration than DIANA. Also, the focus in DIANA is on UpCom and its DownCom step is the same as in GD, with the full model broadcast at every iteration, so that its TotalCom complexity can be *worse* than the one of GD. Extensions of DIANA with bidirectional CC, i.e. compression in both UpCom and DownCom, have been proposed (Gorbunov et al., 2020b; Philippenko & Dieuleveut, 2020; Liu et al., 2020; Condat & Richtárik, 2022), but this does not improve its TotalCom complexity; see also Philippenko & Dieuleveut (2021) and references therein on bidirectional CC. We note that if LT is disabled ($p = 1$), CompressedScaffnew is still new and does not revert to a known algorithm with CC.

## 2 GOALS, CHALLENGES, CONTRIBUTIONS

Our new algorithm CompressedScaffnew builds upon the LT mechanism of Scaffnew and enables CC. In short,

$$\text{CompressedScaffnew} = \underbrace{\text{GD} + \text{LT}}_{\text{Scaffnew}} + \text{CC}.$$

We focus on the strongly convex setting but we also prove sublinear convergence of CompressedScaffnew in the merely convex case in the Appendix. We emphasize that the problem can be arbitrarily heterogeneous: we don't make any assumption on the functions $f_i$ beyond smoothness and strong convexity, and there is no notion of data similarity whatsoever. We also stress that our goal is to deepen our theoretical understanding of LT and CC, and to make these two intuitive and effective mechanisms, which are widely used in practice, work in the best possible way when harnessed to GD. It would certainly be interesting to consider possibly variance-reduced (Gorbunov et al., 2020a; Gower et al., 2020) SGD local steps, as was done for Scaffnew in Malinovsky et al. (2022). We leave it for future work, since we focus on the *communication* complexity, and stochastic gradients can only *worsen* it. Reducing the *computation* complexity using accelerated (Nesterov, 2004) or stochastic GD steps is somewhat orthogonal to our present study.

It is very challenging to combine LT and CC. In the strongly convex and heterogeneous case considered here, the methods Qsparse-local-SGD (Basu et al., 2020) and FedPAQ (Reisizadeh et al., 2020) do not converge linearly. The only linearly converging LT + CC algorithm we are aware of is FedCOMGATE (Haddadpour et al., 2021). But its rate is $\mathcal{O}(d\kappa \log \epsilon^{-1})$, which does not show any acceleration. By contrast, our algorithm is the first, to the best of our knowledge, to exhibit a doubly-accelerated linear rate, by leveraging LT and CC. We note that random reshuffling, which can be seen as a kind of LT, has been combined with CC in Sadiev et al. (2022). The TotalCom complexity of the various algorithms is reported in Table 1.

The program of combining LT and CC looks simple, but the naive approach of plugging compressors into Scaffnew does not work. The key of our design is to combine the two stochastic processes of probabilistic communication and compression with a random mask *in two different ways*, for updating after communication the model estimates $x_i$ on one hand, and the control variates $h_i$ on the other hand. Indeed, a crucial property is that the sum of the control variates over all clients always remains zero. If there is no compression ($s = n$), the two mechanisms coincide, since there is only one source of randomness, which is the coin flip to trigger communication, and CompressedScaffnew reverts to Scaffnew. Our approach relies on a dedicated design of the compressors, explained in Figure 1, so that the messages sent by the different clients complement each other, to keep a tight control of the variance after aggregation. We currently don't know how to use any other type of compressors in CompressedScaffnew.

---

**Algorithm 1** CompressedScaffnew

1: **input:** stepsizes $\gamma > 0$, $\eta > 0$, probability $p \in (0,1]$, $s \in \{2, \ldots, n\}$, initial iterates $x_1^0, \ldots, x_n^0 \in \mathbb{R}^d$, initial control variates $h_1^0, \ldots, h_n^0 \in \mathbb{R}^d$ such that $\sum_{i=1}^n h_i^0 = 0$, sequence of independent coin flips $\theta^0, \theta^1, \ldots$ with $\mathrm{Prob}(\theta^t = 1) = p$, known by the server and all clients, and for every $t$ with $\theta^t = 1$, a random binary mask $\mathbf{q}^t = (q_i^t)_{i=1}^n \in \mathbb{R}^{d \times n}$ with $s$ ones per row, generated as explained in Figure 1, so that the binary vector $q_i^t \in \mathbb{R}^d$ is known by client $i$ and the server. The compressed vector $\mathcal{C}_i^t(v)$ is $v$ multiplied elementwise by $q_i^t$.
2: **for** $t = 0, 1, \ldots$ **do**
3:     **for** $i = 1, \ldots, n$, at clients in parallel, **do**
4:         $\hat{x}_i^t := x_i^t - \gamma \nabla f_i(x_i^t) + \gamma h_i^t$
5:         **if** $\theta^t = 1$ **then**
6:             send $\mathcal{C}_i^t(\hat{x}_i^t)$ to the server, which aggregates and broadcasts $\bar{x}^t := \frac{1}{s} \sum_{j=1}^n \mathcal{C}_j^t(\hat{x}_j^t)$
7:             $x_i^{t+1} := \bar{x}^t$
8:             $h_i^{t+1} := h_i^t + \frac{p\eta}{\gamma}\left(\mathcal{C}_i^t(\bar{x}^t) - \mathcal{C}_i^t(\hat{x}_i^t)\right)$
9:         **else**
10:           $x_i^{t+1} := \hat{x}_i^t$
11:           $h_i^{t+1} := h_i^t$
12:         **end if**
13:     **end for**
14: **end for**

---

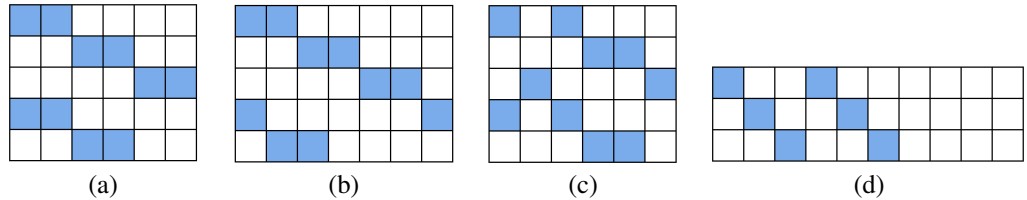

(a)        (b)        (c)        (d)

Figure 1: The random sampling pattern $\mathbf{q}^t = (q_i^t)_{i=1}^n \in \mathbb{R}^{d \times n}$ used for communication is generated by a random permutation of the columns of a fixed binary template pattern, which has the prescribed number $s \geq 2$ of ones in every row. In (a) with $(d, n, s) = (5, 6, 2)$ and (b) with $(d, n, s) = (5, 7, 2)$, with ones in blue and zeros in white, examples of the template pattern used when $d \geq \frac{n}{s}$: for every row $k \in [d]$, there are $s$ ones at columns $i = \mathrm{mod}(s(k-1), n) + 1, \ldots, \mathrm{mod}(sk - 1, n) + 1$. Thus, there are $\lfloor \frac{sd}{n} \rfloor$ or $\lceil \frac{sd}{n} \rceil$ ones in every column vector $q_i$. In (c), an example of sampling pattern obtained after a permutation of the columns of the template pattern in (a). In (d) with $(d, n, s) = (3, 10, 2)$, an example of the template pattern used when $\frac{n}{s} \geq d$: for every column $i = 1, \ldots, ds$, there is 1 one at row $k = \mathrm{mod}(i - 1, d) + 1$. Thus, there is 0 or 1 one in every column vector $q_i$. We can note that when $d = \frac{n}{s}$, the two different rules for $d \geq \frac{n}{s}$ and $\frac{n}{s} \geq d$ for constructing the template pattern are equivalent, since they give exactly the same set of sampling patterns when permuting their columns. These two rules make it possible to generate easily the columns $q_i^t$ of $\mathbf{q}^t$ on the fly, without having to generate the whole mask $\mathbf{q}^t$ explicitly.

Thus, by making use of CC on top of LT, CompressedScaffnew establishes the new state of the art in communication efficiency. If $c$ is small and $n$ is large, its TotalCom complexity is

$$\mathcal{O}\left(\left(\sqrt{d}\sqrt{\kappa} + d\right) \log \epsilon^{-1}\right);$$

our general result is in Theorem 3. Thus, CompressedScaffnew enjoys twofold acceleration, with $\sqrt{\kappa}$ instead of $\kappa$ thanks to LT and $\sqrt{d}$ instead of $d$ thanks to CC.

## 3   Proposed algorithm CompressedScaffnew

The proposed algorithm CompressedScaffnew is shown as Algorithm 1. At every iteration $t \geq 0$, every client $i \in [n]$ performs a gradient descent step with respect to its private cost $f_i$, evaluated at its local model $x_i^t$, with a correction term by its control variate $h_i^t$. This yields a prediction $\hat{x}_i^t$

Table 1: TotalCom complexity of linearly converging algorithms using Local Training (LT), Communication Compression (CC), or both. The $\widetilde{\mathcal{O}}$ notation hides the $\log \epsilon^{-1}$ factor. We note that some of the referenced methods incorporate features of practical interest like the use of stochastic gradients or partial participation. They are irrelevant for our purpose, since they can only *worsen* the communication complexity, which is the object of our study.

| Algorithm | LT | CC | TotalCom | TotalCom=UpCom when $c=0$ |
|---|---|---|---|---|
| DIANA [a] | ✗ | ✓ | $\widetilde{\mathcal{O}}\left((1+cd+\frac{d+cd^2}{n})\kappa + d + cd^2\right)$ | $\widetilde{\mathcal{O}}\left((1+\frac{d}{n})\kappa + d\right)$ |
| EF21 [b] | ✗ | ✓ | $\widetilde{\mathcal{O}}(d\kappa)$ | $\widetilde{\mathcal{O}}(d\kappa)$ |
| Scaffold | ✓ | ✗ | $\widetilde{\mathcal{O}}(d\kappa)$ | $\widetilde{\mathcal{O}}(d\kappa)$ |
| FedLin | ✓ | ✗ | $\widetilde{\mathcal{O}}(d\kappa)$ | $\widetilde{\mathcal{O}}(d\kappa)$ |
| S-Local-GD | ✓ | ✗ | $\widetilde{\mathcal{O}}(d\kappa)$ | $\widetilde{\mathcal{O}}(d\kappa)$ |
| Scaffnew | ✓ | ✗ | $\widetilde{\mathcal{O}}(d\sqrt{\kappa})$ | $\widetilde{\mathcal{O}}(d\sqrt{\kappa})$ |
| FedCOMGATE | ✓ | ✓ | $\widetilde{\mathcal{O}}(d\kappa)$ | $\widetilde{\mathcal{O}}(d\kappa)$ |
| CompressedScaffnew | ✓ | ✓ | $\widetilde{\mathcal{O}}\left(d\frac{\sqrt{\kappa}}{\sqrt{n}}+\sqrt{d}\sqrt{\kappa}+d+\sqrt{c}\,d\sqrt{\kappa}\right)$ | $\widetilde{\mathcal{O}}\left(d\frac{\sqrt{\kappa}}{\sqrt{n}}+\sqrt{d}\sqrt{\kappa}+d\right)$ |

($a$) using independent `rand-1` compressors, for instance. Note that $\mathcal{O}(\sqrt{d}\sqrt{\kappa} + d)$ is better than $\mathcal{O}(\kappa + d)$ and $\mathcal{O}(d\frac{\sqrt{\kappa}}{\sqrt{n}} + d)$ is better than $\mathcal{O}(\frac{d}{n}\kappa + d)$, so that CompressedScaffnew has a better complexity than DIANA.
($b$) using `top-k` compressors with any $k$, for instance.

of the updated local model. Then a random coin flip is made, to decide whether communication occurs or not. Communication occurs with probability $p \in (0, 1]$, with $p$ typically small. If there is no communication, $x_i^{t+1}$ is simply set as $\hat{x}_i^t$ and $h_i$ is unchanged. If communication occurs, every client sends a compressed version of $\hat{x}_i^t$; that is, it sends only a few of its elements, selected randomly according to the rule explained in Figure 1 and known by both the clients and the server (for decoding). The server aggregates the received vectors and forms $\bar{x}^t$, which is broadcast to all clients. They all resume with this fresh estimate of the solution. Every client updates its control variates $h_i$ by modifying only the coordinates which have been involved in the communication process; that is, for which $q_i^t$ has a one. Indeed, the other coordinates of $\hat{x}_i^t$ have not participated to the formation of $\bar{x}^t$, so the received vector $\bar{x}^t$ does not contain relevant information to update $h_i^t$ at these coordinates.

The probability $p \in (0, 1]$ of communication controls the amount of LT, since the expected number of local GD steps between two successive communication rounds is $1/p$. If $p = 1$, communication happens at every iteration and LT is disabled. The sparsity index $s \in \{2, \ldots, n\}$ controls the amount of compression: the lower $s$, the more compression. If $s = n$ and $\eta = 1$, there is no compression and $\mathcal{C}_i^t(v) = v$ for any $v$; then CompressedScaffnew reverts to Scaffnew.

Our main result, stating linear convergence of CompressedScaffnew to the exact solution $x^\star$ of equation 1, is the following:

**Theorem 1.** *In CompressedScaffnew, suppose that*

$$0 < \gamma < \frac{2}{L} \quad and \quad 0 < \eta \leq \frac{n(s-1)}{s(n-1)} \in \left(\frac{1}{2}, 1\right]. \tag{3}$$

*For every $t \geq 0$, define the Lyapunov function*

$$\Psi^t := \frac{1}{\gamma}\sum_{i=1}^{n}\left\|x_i^t - x^\star\right\|^2 + \frac{\gamma}{p^2\eta}\frac{n-1}{s-1}\sum_{i=1}^{n}\left\|h_i^t - h_i^\star\right\|^2, \tag{4}$$

*where $x^\star$ is the unique solution to equation 1 and $h_i^\star = \nabla f_i(x^\star)$. Then CompressedScaffnew converges linearly: for every $t \geq 0$,*

$$\mathbb{E}\left[\Psi^t\right] \leq \rho^t \Psi^0, \tag{5}$$

*where*

$$\rho := \max\left((1-\gamma\mu)^2, (\gamma L - 1)^2, 1 - p^2\eta\frac{s-1}{n-1}\right) < 1. \tag{6}$$

*Also, for every $i \in [n]$, $(x_i^t)_{t\in\mathbb{N}}$ and $(\hat{x}_i^t)_{t\in\mathbb{N}}$ both converge to $x^\star$ and $(h_i^t)_{t\in\mathbb{N}}$ converges to $h_i^\star$, almost surely.*

**Remark 1.** *One can simply set $\eta = \frac{1}{2}$ in* CompressedScaffnew, *which is independent of $n$ and $s$. However, the larger $\eta$, the better, so it is recommended to set*

$$\eta = \frac{n(s-1)}{s(n-1)}. \tag{7}$$

### 3.1 ITERATION COMPLEXITY

CompressedScaffnew has the same iteration complexity as GD, with rate $\rho^{\sharp} := \max(1-\gamma\mu, \gamma L-1)^2$, as long as $p$ and $s$ are large enough to have

$$1 - p^2 \eta \frac{s-1}{n-1} \leq \rho^{\sharp}.$$

This is remarkable: compression during aggregation with $p < 1$ and $s < n$ does not harm convergence at all, until some threshold. This is in contrast with other algorithms with CC, like DIANA, where even a small amount of compression worsens the worst-case complexity.

For any $s \geq 2$, $p \in (0,1]$, $\gamma = \Theta(\frac{1}{L})$, and fixed $\eta \in (0,1]$, the asymptotic iteration complexity of CompressedScaffnew to reach $\epsilon$-accuracy, i.e. $\mathbb{E}[\Psi^t] \leq \epsilon$, is

$$\mathcal{O}\left(\left(\kappa + \frac{n}{sp^2}\right) \log \epsilon^{-1}\right). \tag{8}$$

Thus, by choosing

$$p = \min\left(\sqrt{\frac{(1-\rho^{\sharp})(n-1)}{\eta(s-1)}}, 1\right), \tag{9}$$

or more generally

$$p = \min\left(\Theta\left(\sqrt{\frac{n}{s\kappa}}\right), 1\right), \tag{10}$$

the iteration complexity becomes

$$\mathcal{O}\left(\left(\kappa + \frac{n}{s}\right) \log \epsilon^{-1}\right).$$

In particular, with the choice recommended in equation 13 of $s = \max(2, \lfloor \frac{n}{d} \rfloor, \lfloor cn \rfloor)$, which yields the best TotalCom complexity, the iteration complexity is

$$\mathcal{O}\left(\left(\kappa + \min\left(d, n, \frac{1}{c}\right)\right) \log \epsilon^{-1}\right)$$

(with $\frac{1}{c} = +\infty$ if $c = 0$).

### 3.2 CONVERGENCE IN THE CONVEX CASE

In this section only, we remove the hypothesis of strong convexity: the functions $f_i$ are just assumed to be convex and $L$-smooth, and we suppose that a solution $x^\star \in \mathbb{R}^d$ to equation 1 exists. Then we have sublinear ergodic convergence:

**Theorem 2.** *In* CompressedScaffnew, *suppose that*

$$0 < \gamma < \frac{2}{L} \quad \text{and} \quad 0 < \eta < \frac{n(s-1)}{s(n-1)} \in \left(\frac{1}{2}, 1\right]. \tag{11}$$

*Then, for every $i = 1, \ldots, n$ and $T \geq 0$,*

$$\mathbb{E}\left[\left\|\nabla f(\tilde{x}_i^T)\right\|^2\right] = \mathcal{O}\left(\frac{1}{T}\right), \tag{12}$$

*where $\tilde{x}_i^T := \frac{1}{T+1}\sum_{t=0}^{T} x_i^t$ (an explicit upper bound is given in the proof).*

## 4 COMMUNICATION COMPLEXITY

For any $s \geq 2$, $p \in (0,1]$, $\gamma = \Theta(\frac{1}{L})$, and fixed $\eta \in (0,1]$, the asymptotic iteration complexity of CompressedScaffnew is given in equation 8. Communication occurs at every iteration with probability $p$, and during every communication round, DownCom consists in broadcasting the full $d$-dimensional vector $\bar{x}^t$, whereas in UpCom, compression is effective and the number of real values sent in parallel by the clients is equal to the number of ones per column in the sampling pattern $q$, which is $\lceil \frac{sd}{n} \rceil \geq 1$. Hence, the communication complexities are:

$$\text{DownCom:} \quad \mathcal{O}\left( pd\left( \kappa + \frac{n}{sp^2} \right) \log \epsilon^{-1} \right),$$

$$\text{UpCom:} \quad \mathcal{O}\left( p\left( \frac{sd}{n} + 1 \right)\left( \kappa + \frac{n}{sp^2} \right) \log \epsilon^{-1} \right).$$

$$\text{TotalCom:} \quad \mathcal{O}\left( p\left( \frac{sd}{n} + 1 + cd \right)\left( \kappa + \frac{n}{sp^2} \right) \log \epsilon^{-1} \right).$$

For a given $s$, the best choice for $p$, for both DownCom and UpCom, is given in equation 9, or more generally equation 10, for which

$$\mathcal{O}\left( p\left( \kappa + \frac{n}{sp^2} \right) \right) = \mathcal{O}\left( \sqrt{\frac{n\kappa}{s}} + \frac{n}{s} \right)$$

and the TotalCom complexity is

$$\text{TotalCom:} \quad \mathcal{O}\left( \left( \sqrt{\frac{n\kappa}{s}} + \frac{n}{s} \right)\left( \frac{sd}{n} + 1 + cd \right) \log \epsilon^{-1} \right).$$

We see the first acceleration effect due to LT: with a suitable $p < 1$, the communication complexity only depends on $\sqrt{\kappa}$, not $\kappa$, whatever the compression level $s$. Without compression, i.e. $s = n$, CompressedScaffnew reverts to Scaffnew, with TotalCom complexity $\mathcal{O}(d\sqrt{\kappa} \log \epsilon^{-1})$. We can now set $s$ to further accelerate the algorithm, by minimizing the TotalCom complexity:

**Theorem 3.** *In* CompressedScaffnew, *suppose that equation 3 holds, with $\eta$ fixed and $\gamma = \Theta(\frac{1}{L})$, that $p$ satisfies equation 10 and that*

$$s = \max\left( 2, \left\lfloor \frac{n}{d} \right\rfloor, \lfloor cn \rfloor \right). \tag{13}$$

*Then the TotalCom complexity of* CompressedScaffnew *is*

$$\mathcal{O}\left( \left( d\frac{\sqrt{\kappa}}{\sqrt{n}} + \sqrt{d}\sqrt{\kappa} + d + \sqrt{c}\,d\sqrt{\kappa} \right) \log \epsilon^{-1} \right). \tag{14}$$

Hence, as long as $c \leq \max(\frac{2}{n}, \frac{1}{d}, \frac{1}{\kappa})$, there is no difference with the case $c = 0$, in which we only focus on UpCom, and the TotalCom complexity is

$$\mathcal{O}\left( \left( d\frac{\sqrt{\kappa}}{\sqrt{n}} + \sqrt{d}\sqrt{\kappa} + d \right) \log \epsilon^{-1} \right).$$

On the other hand, if $c \geq \max(\frac{2}{n}, \frac{1}{d}, \frac{1}{\kappa})$, the complexity increases and becomes $\mathcal{O}(\sqrt{c}\,d\sqrt{\kappa} \log \epsilon^{-1})$, but compression remains operational and effective with the $\sqrt{c}$ factor. It is only when $c = 1$ that $s = n$, i.e. there is no compression and CompressedScaffnew reverts to Scaffnew, and that the Upcom, DownCom and TotalCom complexities all become $\mathcal{O}(d\sqrt{\kappa} \log \epsilon^{-1})$. In any case, for every $c \in [0,1]$, CompressedScaffnew is faster than Scaffnew.

We have reported in Table 1 the TotalCom complexity for several algorithms, and to the best of our knowledge, CompressedScaffnew improves upon all known algorithms, which use either LT or CC on top of GD. Moreover, this is achieved not only for uplink communication, but for our more comprehensive model of total communication.

Carrying out large-scale experiments is beyond the scope of this work, which studies the foundational algorithmic and theoretical properties of a class of algorithms. Nevertheless, we illustrate and confirm our results on a practical logistic regression problem. The results are shown in Figures 2 and 3 and we refer to the Appendix for the explanations, by lack of space.

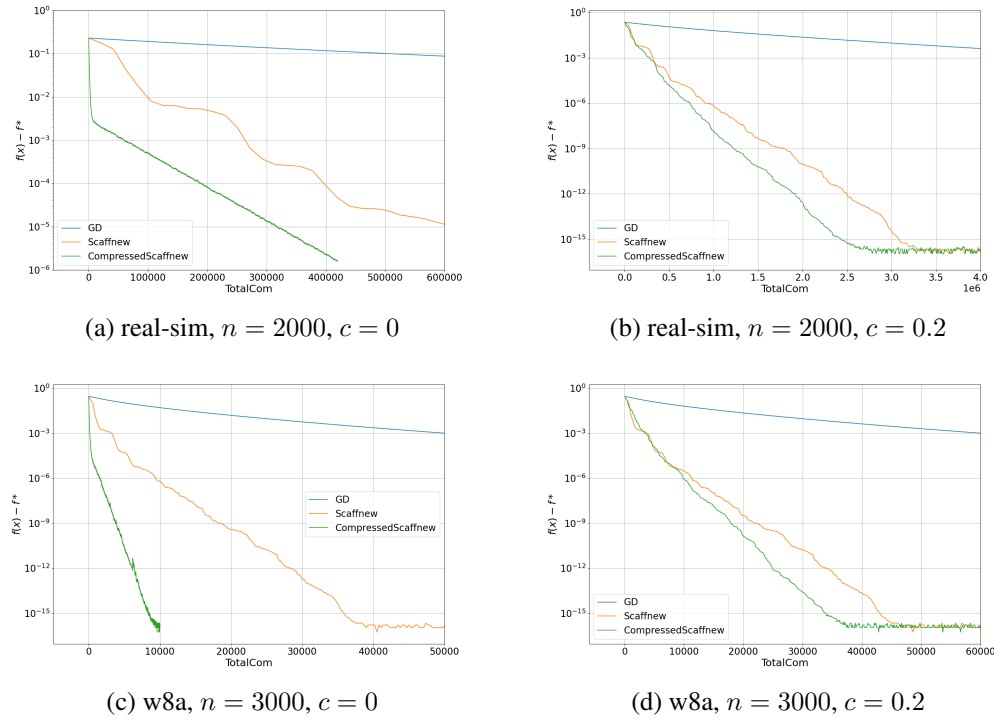

Figure 2: Logistic regression experiment. The datasets real-sim and w8a have $d = 20,958$ and $d = 300$ features, respectively. In (a) and (b), $d \approx 10n$, whereas in (c) and (d), this is the opposite with $n = 10d$.

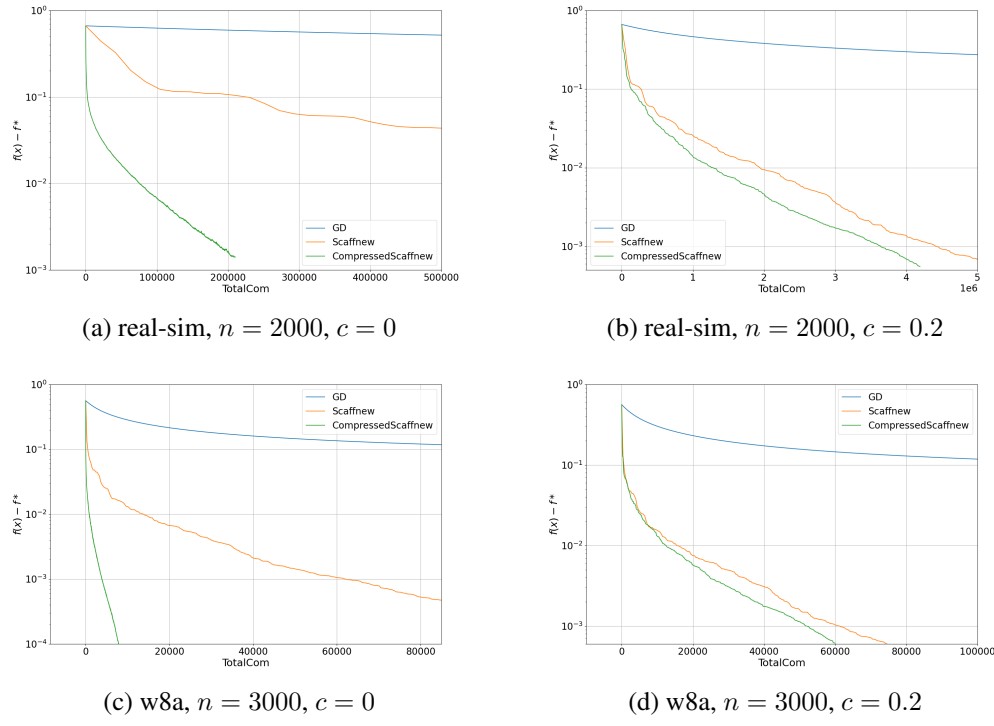

Figure 3: Logistic regression experiment. The setting is the same as in Figure 2, but with $\kappa = 10^6$ instead of 334.

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

# Appendix

## CONTENTS

Table 2: Summary of the main notations used in the paper.

| | |
|---|---|
| LT | local training |
| CC | communication compression |
| $L$ | smoothness constant |
| $\mu$ | strong convexity constant |
| $\kappa = L/\mu$ | condition number of the functions |
| $d$ | dimension of the model |
| $n, i$ | number and index of clients |
| $[n] = \{1, \ldots, n\}$ | |
| $c$ | weight on downlink communication (DownCom), see equation 2 |
| $s \in \{2, \ldots, n\}$ | sparsity index for compression. No compression if $s = n$ |
| $\mathbf{q} = (q_i)_{i=1}^c$ | random binary mask for compression, as detailed in Figure 1 |
| $p$ | probability that communication occurs at any iteration |
| $p^{-1}$ | expected number of local steps per round |
| $t, T$ | indexes of iterations |
| $\gamma, \eta, \tau$ | stepsizes |
| $x_i$ | local model estimate at client $i$ |
| $h_i$ | local control variate tracking $\nabla f_i$ |
| $\bar{x}$ | model estimate at the server |
| $\rho$ | convergence rate |

## A    EXPERIMENTS

We consider a logistic regression problem. The global loss function is

$$f(x) = \frac{1}{M} \sum_{m=1}^{M} \log(1 + \exp(-b_m a_m^\top x)) + \frac{\mu}{2}\|x\|^2, \tag{15}$$

where the $a_m \in \mathbb{R}^d$ and $b_m \in \{-1, 1\}$ are data samples and $M$ is their total number. The function $f$ in equation 15 is split into $n$ functions $f_i$ (the remainder of $M$ divided by $n$ samples is discarded). The strong convexity parameter $\mu$ is set to $0.003L_0$ in Figure 2 and $10^{-6}L_0$ in Figure 3, where $L_0$ is the smoothness constant without the $\frac{\mu}{2}\|x\|^2$ term (so that $L = L_0 + \mu$). We consider the case where the number of clients is larger than the model dimension ($n > d$) and vice versa. For this, we use the 'w8a' and 'real-sim' datasets from the classical LIBSVM library (Chang & Lin, 2011). For each of them, we consider the two cases $c = 0$ and $c = 0.2$.

We measure the convergence error $f(x) - f(x^\star)$ with respect to the TotalCom amount of communication, where $x$ is $x^t$ for GD and $\bar{x}^t$ when communication occurs for CompressedScaffnew and Scaffnew. The objective gap $f(x) - f(x^\star)$ is a fair way to compare different algorithms and, since $f$ is $L$-smooth, $f(x) - f(x^\star) \leq \frac{L}{2}\|x - x^\star\|^2$ for any $x$, so that it is guaranteed to converge linearly with the same rate as $\Psi$ in Theorem 1.

The stepsize $\gamma = \frac{2}{L+\mu}$ is used in all algorithms. The probability $p$ is set to $\frac{1}{\sqrt{\kappa}}$ for Scaffnew and $\min(\sqrt{\frac{n}{s\kappa}}, 1)$ for CompressedScaffnew, with $s$ and $\eta$ set according to equation 13 and equation 7, respectively. $x^0$ and the $h_i^0$ are all set to zero vectors.

The results are shown in Figures 2 and 3. The algorithms converge linearly, and CompressedScaffnew is faster than Scaffnew, as expected. This confirms that the proposed compression technique is effective. The speedup of CompressedScaffnew over Scaffnew is higher for $c = 0$ than for $c = 0.2$. This is also expected, since when $c$ increases, there is less compression and the two algorithms become more similar; they are the same, without compression, when $c = 1$.

## B    CONCLUSION

We have proposed CompressedScaffnew, the first communication-efficient algorithm for distributed optimization that provably benefits from the two combined acceleration mechanisms of Local Training (LT) and Communication Compression (CC). Moreover, this is achieved not only for uplink

communication, but for our more comprehensive model of total communication. These theoretical guarantees are confirmed in practice and CompressedScaffnew communicates less than existing algorithms to reach the same accuracy. An important venue for future work will be to generalize our specific compression mechanism to a broad class of compressors including quantization (Horváth et al., 2022). Also, bidirectional compression, i.e. applying compression to not only uplink but also downlink communication, should be investigated (Liu et al., 2020; Philippenko & Dieuleveut, 2021). Another venue consists in replacing the true gradients by stochastic estimates, in combination with variance reduction strategies, as was done for Scaffnew in Malinovsky et al. (2022). Analyzing the properties of CompressedScaffnew on nonconvex problems should also be studied (Karimireddy et al., 2021; Das et al., 2022).

## C   PROOF OF THEOREM 1

We introduce vector notations to simplify the derivations: the problem equation 1 can be written as

$$\text{find } \mathbf{x}^\star = \arg\min_{\mathbf{x} \in \mathcal{X}} \mathbf{f}(\mathbf{x}) \quad \text{s.t.} \quad W\mathbf{x} = 0, \tag{16}$$

where $\mathcal{X} := \mathbb{R}^{d \times n}$, an element $\mathbf{x} = (x_i)_{i=1}^n \in \mathcal{X}$ is a collection of vectors $x_i \in \mathbb{R}^d$, $\mathbf{f} : \mathbf{x} \in \mathcal{X} \mapsto \sum_{i=1}^n f_i(x_i)$ is $L$-smooth and $\mu$-strongly convex, the linear operator $W : \mathcal{X} \to \mathcal{X}$ maps $\mathbf{x} = (x_i)_{i=1}^n$ to $(x_i - \frac{1}{n} \sum_{j=1}^n x_j)_{i=1}^n$. The constraint $W\mathbf{x} = 0$ means that $\mathbf{x}$ minus its average is zero; that is, $\mathbf{x}$ has identical components $x_1 = \cdots = x_n$. Thus, equation 16 is indeed equivalent to equation 1. We have $W = W^* = W^2$.

We solve the problem equation 16 using the following algorithm, which will be shown below to be CompressedScaffnew:

---

**Algorithm 2**

---

**input:** stepsizes $\gamma > 0, \tau > 0$; $s \in \{2, \ldots, n\}$; initial estimates $\mathbf{x}^0 \in \mathcal{X}, \mathbf{u}^0 \in \mathcal{X}$ with $\sum_{i=1}^n u_i^0 = 0$; constant $\omega \geq 0$; sequence of independent coin flips $\theta^0, \theta^1, \ldots$ with $\text{Prob}(\theta^t = 1) = p$, and for every $t$ with $\theta^t = 1$, a random binary mask $\mathbf{q}^t = (q_i^t)_{i=1}^n \in \mathbb{R}^{d \times n}$ generated as explained in Figure 1. The compressed vector $\mathcal{C}_i^t(v)$ is $v$ multiplied elementwise by $q_i^t$.
**for** $t = 0, 1, \ldots$ **do**
    $\hat{\mathbf{x}}^t := \mathbf{x}^t - \gamma \nabla \mathbf{f}(\mathbf{x}^t) - \gamma \mathbf{u}^t$
    **if** $\theta^t = 1$ **then**
        $\bar{x}^t := \frac{1}{s} \sum_{j=1}^n \mathcal{C}_j^t(\hat{x}_j^t)$
        $\mathbf{x}^{t+1} := \bar{\mathbf{x}}^t$, with $\bar{x}_i^t = \bar{x}^t$, for every $i = 1, \ldots, n$
    **else**
        $\mathbf{x}^{t+1} := \hat{\mathbf{x}}^t$
    **end if**
    $\mathbf{d}^t :\approx W\hat{\mathbf{x}}^t$
    $\mathbf{u}^{t+1} := \mathbf{u}^t + \frac{\tau}{1+\omega} \mathbf{d}^t$
**end for**

---

We denote by $\mathcal{F}_t$ the $\sigma$-algebra generated by the collection of $\mathcal{X}$-valued random variables $(\mathbf{x}^0, \mathbf{u}^0), \ldots, (\mathbf{x}^t, \mathbf{u}^t)$, for every $t \geq 0$. In Algorithm 2, $\mathbf{d}^t :\approx W\hat{\mathbf{x}}^t$ means that $\mathbf{d}^t$ is a random variable with expectation $W\hat{\mathbf{x}}^t$. Its construction, so that Algorithm 2 becomes CompressedScaffnew, is explained in Section C.1, but the convergence analysis of Algorithm 2 only relies on the 3 following properties of this stochastic process, which are supposed to hold: for every $t \geq 0$,

1. $\mathbb{E}[\mathbf{d}^t \mid \mathcal{F}^t] = W\hat{\mathbf{x}}^t$.

2. There exists a value $\omega \geq 0$ such that

$$\mathbb{E}\left[ \left\| \mathbf{d}^t - W\hat{\mathbf{x}}^t \right\|^2 \mid \mathcal{F}^t \right] \leq \omega \left\| W\hat{\mathbf{x}}^t \right\|^2. \tag{17}$$

3. $\mathbf{d}^t$ belongs to the range of $W$; that is, $\sum_{i=1}^n d_i^t = 0$.

In Algorithm 2, we suppose that $\sum_{i=1}^{n} u_i^0 = 0$. Then, it follows from the third property of $\mathbf{d}^t$ that, for every $t \geq 0$, $\sum_{i=1}^{n} u_i^t = 0$; that is, $W\mathbf{u}^t = \mathbf{u}^t$.

Algorithm 2 converges linearly:

**Theorem 4.** *In Algorithm 2, suppose that $0 < \gamma < \frac{2}{L}$ and that $\tau \leq \frac{p}{\gamma} \frac{n(s-1)}{s(n-1)}$. For every $t \geq 0$, define the Lyapunov function*

$$\Psi^t := \frac{1}{\gamma} \left\| \mathbf{x}^t - \mathbf{x}^\star \right\|^2 + \frac{1+\omega}{\tau} \left\| \mathbf{u}^t - \mathbf{u}^\star \right\|^2, \tag{18}$$

*where $\mathbf{x}^\star$ is the unique solution to equation 16 and $\mathbf{u}^\star := -\nabla\mathbf{f}(\mathbf{x}^\star)$. Then Algorithm 2 converges linearly: for every $t \geq 0$,*

$$\mathbb{E}\left[\Psi^t\right] \leq \rho^t \Psi^0, \tag{19}$$

*where*

$$\rho := \max\left( (1 - \gamma\mu)^2, (\gamma L - 1)^2, 1 - \frac{\gamma\tau}{1+\omega} \right) < 1. \tag{20}$$

*Also, $(\mathbf{x}^t)_{t \in \mathbb{N}}$ and $(\hat{\mathbf{x}}^t)_{t \in \mathbb{N}}$ both converge to $\mathbf{x}^\star$ and $(\mathbf{u}^t)_{t \in \mathbb{N}}$ converges to $\mathbf{u}^\star$, almost surely.*

*Proof.* For every $t \geq 0$, we define $\hat{\mathbf{u}}^{t+1} := \mathbf{u}^t + \tau W\hat{\mathbf{x}}^t$, $\mathbf{w}^t := \mathbf{x}^t - \gamma\nabla\mathbf{f}(\mathbf{x}^t)$ and $\mathbf{w}^\star := \mathbf{x}^\star - \gamma\nabla\mathbf{f}(\mathbf{x}^\star)$. We also define $\bar{\mathbf{x}}^{t\sharp} := (\bar{x}^{t\sharp})_{i=1}^n$, with $\bar{x}^{t\sharp} := \frac{1}{n}\sum_{i=1}^n \hat{x}_i^t$; that is, $\bar{\mathbf{x}}^{t\sharp}$ is the exact average of the $\hat{x}_i^t$, of which $\bar{x}^t$ is an unbiased random estimate.

We have

$$\mathbb{E}\left[\left\|\mathbf{x}^{t+1} - \mathbf{x}^\star\right\|^2 \mid \mathcal{F}_t\right] = p\mathbb{E}_{\mathbf{q}^t}\left[\left\|\bar{\mathbf{x}}^t - \mathbf{x}^\star\right\|^2 \mid \mathcal{F}_t\right] + (1-p)\left\|\hat{\mathbf{x}}^t - \mathbf{x}^\star\right\|^2,$$

where $\mathbb{E}_{\mathbf{q}^t}$ denotes the expectation with respect to the random mask $\mathbf{q}^t$. To analyze $\mathbb{E}_{\mathbf{q}^t}\left[\left\|\bar{\mathbf{x}}^t - \mathbf{x}^\star\right\|^2 \mid \mathcal{F}_t\right]$, we can remark that the expectation and the squared Euclidean norm are separable with respect to the coordinates of the $d$-dimensional vectors, so that we can reason on the coordinates independently on each other, even if the the coordinates, or rows, of $\mathbf{q}^t$ are mutually dependent. Thus, for every coordinate $k \in [d]$, it is like a subset $\Omega_k^t \subset [n]$ of size $s$, which corresponds to the location of the ones in the $k$-th row of $\mathbf{q}^t$, is chosen uniformly at random and

$$\bar{x}_k^t = \frac{1}{s} \sum_{i \in \Omega_k^t} \hat{x}_{i,k}^t.$$

Since $\mathbb{E}_{\mathbf{q}^t}[\bar{\mathbf{x}}^t \mid \mathcal{F}_t] = \bar{\mathbf{x}}^{t\sharp}$,

$$\mathbb{E}_{\mathbf{q}^t}\left[\left\|\bar{\mathbf{x}}^t - \mathbf{x}^\star\right\|^2 \mid \mathcal{F}_t\right] = \left\|\bar{\mathbf{x}}^{t\sharp} - \mathbf{x}^\star\right\|^2 + \mathbb{E}_{\mathbf{q}^t}\left[\left\|\bar{\mathbf{x}}^t - \bar{\mathbf{x}}^{t\sharp}\right\|^2 \mid \mathcal{F}_t\right],$$

with

$$\left\|\bar{\mathbf{x}}^{t\sharp} - \mathbf{x}^\star\right\|^2 = \left\|\hat{\mathbf{x}}^t - \mathbf{x}^\star\right\|^2 - \left\|W\hat{\mathbf{x}}^t\right\|^2$$

and, as proved in Condat & Richtárik (2022, Proposition 1),

$$\mathbb{E}_{\mathbf{q}^t}\left[\left\|\bar{\mathbf{x}}^t - \bar{\mathbf{x}}^{t\sharp}\right\|^2 \mid \mathcal{F}_t\right] = n\sum_{k=1}^d \mathbb{E}_{\Omega_k^t}\left[\left(\frac{1}{s}\sum_{i \in \Omega_k^t} \hat{x}_{i,k}^t - \frac{1}{n}\sum_{j=1}^n \hat{x}_{j,k}^t\right)^2 \mid \mathcal{F}_t\right] = \nu\left\|W\hat{\mathbf{x}}^t\right\|^2,$$

where

$$\nu := \frac{n-s}{s(n-1)} \in \left[0, \frac{1}{2}\right). \tag{21}$$

Moreover,

$$
\begin{aligned}
\left\|\hat{\mathbf{x}}^t - \mathbf{x}^\star\right\|^2 &= \left\|\mathbf{w}^t - \mathbf{w}^\star\right\|^2 + \gamma^2\left\|\mathbf{u}^t - \mathbf{u}^\star\right\|^2 - 2\gamma\langle\mathbf{w}^t - \mathbf{w}^\star, \mathbf{u}^t - \mathbf{u}^\star\rangle \\
&= \left\|\mathbf{w}^t - \mathbf{w}^\star\right\|^2 - \gamma^2\left\|\mathbf{u}^t - \mathbf{u}^\star\right\|^2 - 2\gamma\langle\hat{\mathbf{x}}^t - \mathbf{x}^\star, \mathbf{u}^t - \mathbf{u}^\star\rangle \\
&= \left\|\mathbf{w}^t - \mathbf{w}^\star\right\|^2 - \gamma^2\left\|\mathbf{u}^t - \mathbf{u}^\star\right\|^2 - 2\gamma\langle\hat{\mathbf{x}}^t - \mathbf{x}^\star, \hat{\mathbf{u}}^{t+1} - \mathbf{u}^\star\rangle + 2\gamma\langle\hat{\mathbf{x}}^t - \mathbf{x}^\star, \hat{\mathbf{u}}^{t+1} - \mathbf{u}^t\rangle \\
&= \left\|\mathbf{w}^t - \mathbf{w}^\star\right\|^2 - \gamma^2\left\|\mathbf{u}^t - \mathbf{u}^\star\right\|^2 - 2\gamma\langle\hat{\mathbf{x}}^t - \mathbf{x}^\star, \hat{\mathbf{u}}^{t+1} - \mathbf{u}^\star\rangle + 2\gamma\tau\langle\hat{\mathbf{x}}^t - \mathbf{x}^\star, W\hat{\mathbf{x}}^t\rangle \\
&= \left\|\mathbf{w}^t - \mathbf{w}^\star\right\|^2 - \gamma^2\left\|\mathbf{u}^t - \mathbf{u}^\star\right\|^2 - 2\gamma\langle\hat{\mathbf{x}}^t - \mathbf{x}^\star, \hat{\mathbf{u}}^{t+1} - \mathbf{u}^\star\rangle + 2\gamma\tau\left\|W\hat{\mathbf{x}}^t\right\|^2.
\end{aligned}
$$

Hence,

$$
\begin{aligned}
\mathbb{E}\left[\left\|\mathbf{x}^{t+1} - \mathbf{x}^\star\right\|^2 \mid \mathcal{F}_t\right] &= p\left\|\hat{\mathbf{x}}^t - \mathbf{x}^\star\right\|^2 - p\left\|W\hat{\mathbf{x}}^t\right\|^2 + p\nu\left\|W\hat{\mathbf{x}}^t\right\|^2 + (1-p)\left\|\hat{\mathbf{x}}^t - \mathbf{x}^\star\right\|^2 \\
&= \left\|\hat{\mathbf{x}}^t - \mathbf{x}^\star\right\|^2 - p(1-\nu)\left\|W\hat{\mathbf{x}}^t\right\|^2 \\
&= \left\|\mathbf{w}^t - \mathbf{w}^\star\right\|^2 - \gamma^2\left\|\mathbf{u}^t - \mathbf{u}^\star\right\|^2 - 2\gamma\langle\hat{\mathbf{x}}^t - \mathbf{x}^\star, \hat{\mathbf{u}}^{t+1} - \mathbf{u}^\star\rangle \\
&\quad + \left(2\gamma\tau - p(1-\nu)\right)\left\|W\hat{\mathbf{x}}^t\right\|^2.
\end{aligned}
$$

On the other hand,

$$
\begin{aligned}
\mathbb{E}\left[\left\|\mathbf{u}^{t+1} - \mathbf{u}^\star\right\|^2 \mid \mathcal{F}_t\right] &\leq \left\|\mathbf{u}^t - \mathbf{u}^\star + \frac{1}{1+\omega}\left(\hat{\mathbf{u}}^{t+1} - \mathbf{u}^t\right)\right\|^2 + \frac{\omega}{(1+\omega)^2}\left\|\hat{\mathbf{u}}^{t+1} - \mathbf{u}^t\right\|^2 \\
&= \left\|\frac{\omega}{1+\omega}\left(\mathbf{u}^t - \mathbf{u}^\star\right) + \frac{1}{1+\omega}\left(\hat{\mathbf{u}}^{t+1} - \mathbf{u}^\star\right)\right\|^2 + \frac{\omega}{(1+\omega)^2}\left\|\hat{\mathbf{u}}^{t+1} - \mathbf{u}^t\right\|^2 \\
&= \frac{\omega^2}{(1+\omega)^2}\left\|\mathbf{u}^t - \mathbf{u}^\star\right\|^2 + \frac{1}{(1+\omega)^2}\left\|\hat{\mathbf{u}}^{t+1} - \mathbf{u}^\star\right\|^2 \\
&\quad + \frac{2\omega}{(1+\omega)^2}\langle\mathbf{u}^t - \mathbf{u}^\star, \hat{\mathbf{u}}^{t+1} - \mathbf{u}^\star\rangle + \frac{\omega}{(1+\omega)^2}\left\|\hat{\mathbf{u}}^{t+1} - \mathbf{u}^\star\right\|^2 \\
&\quad + \frac{\omega}{(1+\omega)^2}\left\|\mathbf{u}^t - \mathbf{u}^\star\right\|^2 - \frac{2\omega}{(1+\omega)^2}\langle\mathbf{u}^t - \mathbf{u}^\star, \hat{\mathbf{u}}^{t+1} - \mathbf{u}^\star\rangle \\
&= \frac{1}{1+\omega}\left\|\hat{\mathbf{u}}^{t+1} - \mathbf{u}^\star\right\|^2 + \frac{\omega}{1+\omega}\left\|\mathbf{u}^t - \mathbf{u}^\star\right\|^2.
\end{aligned}
$$

Moreover,

$$
\begin{aligned}
\left\|\hat{\mathbf{u}}^{t+1} - \mathbf{u}^\star\right\|^2 &= \left\|(\mathbf{u}^t - \mathbf{u}^\star) + (\hat{\mathbf{u}}^{t+1} - \mathbf{u}^t)\right\|^2 \\
&= \left\|\mathbf{u}^t - \mathbf{u}^\star\right\|^2 + \left\|\hat{\mathbf{u}}^{t+1} - \mathbf{u}^t\right\|^2 + 2\langle\mathbf{u}^t - \mathbf{u}^\star, \hat{\mathbf{u}}^{t+1} - \mathbf{u}^t\rangle \\
&= \left\|\mathbf{u}^t - \mathbf{u}^\star\right\|^2 + 2\langle\hat{\mathbf{u}}^{t+1} - \mathbf{u}^\star, \hat{\mathbf{u}}^{t+1} - \mathbf{u}^t\rangle - \left\|\hat{\mathbf{u}}^{t+1} - \mathbf{u}^t\right\|^2 \\
&= \left\|\mathbf{u}^t - \mathbf{u}^\star\right\|^2 - \left\|\hat{\mathbf{u}}^{t+1} - \mathbf{u}^t\right\|^2 + 2\tau\langle\hat{\mathbf{u}}^{t+1} - \mathbf{u}^\star, W(\hat{\mathbf{x}}^t - \mathbf{x}^\star)\rangle \\
&= \left\|\mathbf{u}^t - \mathbf{u}^\star\right\|^2 - \tau^2\left\|W\hat{\mathbf{x}}^t\right\|^2 + 2\tau\langle W(\hat{\mathbf{u}}^{t+1} - \mathbf{u}^\star), \hat{\mathbf{x}}^t - \mathbf{x}^\star\rangle \\
&= \left\|\mathbf{u}^t - \mathbf{u}^\star\right\|^2 - \tau^2\left\|W\hat{\mathbf{x}}^t\right\|^2 + 2\tau\langle\hat{\mathbf{u}}^{t+1} - \mathbf{u}^\star, \hat{\mathbf{x}}^t - \mathbf{x}^\star\rangle.
\end{aligned}
$$

Hence,

$$
\begin{aligned}
\frac{1}{\gamma}\mathbb{E}\left[\left\|\mathbf{x}^{t+1} - \mathbf{x}^\star\right\|^2 \mid \mathcal{F}_t\right] &+ \frac{1+\omega}{\tau}\mathbb{E}\left[\left\|\mathbf{u}^{t+1} - \mathbf{u}^\star\right\|^2 \mid \mathcal{F}_t\right] \\
&\leq \frac{1}{\gamma}\left\|\mathbf{w}^t - \mathbf{w}^\star\right\|^2 - \gamma\left\|\mathbf{u}^t - \mathbf{u}^\star\right\|^2 + \left(2\tau - \frac{p}{\gamma}(1-\nu)\right)\left\|W\hat{\mathbf{x}}^t\right\|^2 \\
&\quad - 2\langle\hat{\mathbf{x}}^t - \mathbf{x}^\star, \hat{\mathbf{u}}^{t+1} - \mathbf{u}^\star\rangle + \frac{1}{\tau}\left\|\mathbf{u}^t - \mathbf{u}^\star\right\|^2 \\
&\quad - \tau\left\|W\hat{\mathbf{x}}^t\right\|^2 + 2\langle\hat{\mathbf{u}}^{t+1} - \mathbf{u}^\star, \hat{\mathbf{x}}^t - \mathbf{x}^\star\rangle + \frac{\omega}{\tau}\left\|\mathbf{u}^t - \mathbf{u}^\star\right\|^2 \\
&= \frac{1}{\gamma}\left\|\mathbf{w}^t - \mathbf{w}^\star\right\|^2 + \left(\frac{1+\omega}{\tau} - \gamma\right)\left\|\mathbf{u}^t - \mathbf{u}^\star\right\|^2 \\
&\quad + \left(\tau - \frac{p}{\gamma}(1-\nu)\right)\left\|W\hat{\mathbf{x}}^t\right\|^2.
\end{aligned}
\tag{22}
$$

Since we have supposed $\tau - \frac{p}{\gamma}(1-\nu) \leq 0$,

$$
\begin{aligned}
\frac{1}{\gamma}\mathbb{E}\left[\left\|\mathbf{x}^{t+1} - \mathbf{x}^\star\right\|^2 \mid \mathcal{F}_t\right] &+ \frac{1+\omega}{\tau}\mathbb{E}\left[\left\|\mathbf{u}^{t+1} - \mathbf{u}^\star\right\|^2 \mid \mathcal{F}_t\right] \\
&\leq \frac{1}{\gamma}\left\|\mathbf{w}^t - \mathbf{w}^\star\right\|^2 + \frac{1+\omega}{\tau}\left(1 - \frac{\gamma\tau}{1+\omega}\right)\left\|\mathbf{u}^t - \mathbf{u}^\star\right\|^2.
\end{aligned}
$$

According to Condat & Richtárik (2023, Lemma 1),

$$\left\|\mathbf{w}^t - \mathbf{w}^\star\right\|^2 = \left\|(\mathrm{Id} - \gamma\nabla\mathbf{f})\mathbf{x}^t - (\mathrm{Id} - \gamma\nabla\mathbf{f})\mathbf{x}^\star\right\|^2$$
$$\leq \max(1 - \gamma\mu, \gamma L - 1)^2 \left\|\mathbf{x}^t - \mathbf{x}^\star\right\|^2.$$

Therefore,

$$\mathbb{E}\big[\Psi^{t+1} \mid \mathcal{F}_t\big] \leq \max\left((1 - \gamma\mu)^2, (\gamma L - 1)^2, 1 - \frac{\gamma\tau}{1+\omega}\right)\Psi^t. \tag{23}$$

Using the tower rule, we can unroll the recursion in equation 23 to obtain the unconditional expectation of $\Psi^{t+1}$. Moreover, using classical results on supermartingale convergence (Bertsekas, 2015, Proposition A.4.5), it follows from equation 23 that $\Psi^t \to 0$ almost surely. Almost sure convergence of $\mathbf{x}^t$ and $\mathbf{u}^t$ follows. Finally, by Lipschitz continuity of $\nabla\mathbf{f}$, we can upper bound $\|\hat{\mathbf{x}}^t - \mathbf{x}^\star\|^2$ by a linear combination of $\|\mathbf{x}^t - \mathbf{x}^\star\|^2$ and $\|\mathbf{u}^t - \mathbf{u}^\star\|^2$. It follows that $\mathbb{E}\left[\left\|\hat{\mathbf{x}}^t - \mathbf{x}^\star\right\|^2\right] \to 0$ linearly with the same rate $\rho$ and that $\hat{\mathbf{x}}^t \to \mathbf{x}^\star$ almost surely, as well. □

CompressedScaffnew corresponds to Algorithm 2 with $u_i$ replaced by $-h_i$, the randomization strategy for the dual update detailed in Section C.1, the variance factors $\omega$ and $\nu$ defined in equation 26 and equation 21, respectively, and $\tau := \frac{p}{\gamma}\eta$, for some $\eta$ with

$$0 < \eta \leq 1 - \nu = \frac{n(s-1)}{s(n-1)} \in \left(\frac{1}{2}, 1\right].$$

With these substitutions, Theorem 4 yields Theorem 1.

## C.1  THE RANDOM VARIABLE $\mathbf{d}^t$

We define the random variable $\mathbf{d}^t$ used in Algorithm 2, so that it becomes CompressedScaffnew. If $\theta^t = 0$, $\mathbf{d}^t := 0$. If, on the other hand, $\theta^t = 1$, for every coordinate $k \in [d]$, a subset $\Omega_k^t \subset [n]$ of size $s$ is chosen uniformly at random. These sets $(\Omega_k^t)_{k=1}^d$ are mutually dependent, but this does not matter for the derivations, since we can reason on the coordinates separately. Then, for every $k \in [d]$ and $i \in [n]$,

$$d_{i,k}^t := \begin{cases} a\left(\hat{x}_{i,k}^t - \frac{1}{s}\sum_{j\in\Omega_k^t}\hat{x}_{j,k}^t\right) & \text{if } i \in \Omega_k^t, \\ 0 & \text{otherwise,} \end{cases} \tag{24}$$

for some value $a > 0$ to determine. We can check that $\sum_{i=1}^n d_i^t = 0$. We can also note that $\mathbf{d}^t$ depends only on $W\hat{\mathbf{x}}^t$ and not on $\hat{\mathbf{x}}^t$; in particular, if $\hat{x}_1^t = \cdots = \hat{x}_n^t$, $d_i^t = 0$. We have to set $a$ so that $\mathbb{E}[d_i^t] = \hat{x}_i^t - \frac{1}{n}\sum_{j=1}^n \hat{x}_j^t$, where the expectation is with respect to $\theta^t$ and the $\Omega_k^t$ (all expectations in this section are conditional to $\hat{\mathbf{x}}^t$). So, let us calculate this expectation.

Let $k \in [d]$. For every $i \in [n]$,

$$\mathbb{E}\big[d_{i,k}^t\big] = p\frac{s}{n}\left(a\hat{x}_{i,k}^t - \frac{a}{s}\mathbb{E}_{\Omega:i\in\Omega}\left[\sum_{j\in\Omega}\hat{x}_{j,k}^t\right]\right),$$

where $\mathbb{E}_{\Omega:i\in\Omega}$ denotes the expectation with respect to a subset $\Omega \subset [n]$ of size $s$ containing $i$ and chosen uniformly at random. We have

$$\mathbb{E}_{\Omega:i\in\Omega}\left[\sum_{j\in\Omega}\hat{x}_{j,k}^t\right] = \hat{x}_{i,k}^t + \frac{s-1}{n-1}\sum_{j\in[n]\setminus\{i\}}\hat{x}_{j,k}^t = \frac{n-s}{n-1}\hat{x}_{i,k}^t + \frac{s-1}{n-1}\sum_{j=1}^n\hat{x}_{j,k}^t.$$

Hence, for every $i \in [n]$,

$$\mathbb{E}\big[d_{i,k}^t\big] = p\frac{s}{n}\left(a - \frac{a}{s}\frac{n-s}{n-1}\right)\hat{x}_{i,k}^t - p\frac{s}{n}\frac{a}{s}\frac{s-1}{n-1}\sum_{j=1}^n\hat{x}_{j,k}^t.$$

Therefore, by setting

$$a := \frac{n-1}{p(s-1)}, \tag{25}$$

we have, for every $i \in [n]$,

$$\mathbb{E}\big[d_{i,k}^t\big] = p\frac{s}{n}\left(\frac{1}{p}\frac{n-1}{s-1} - \frac{1}{p}\frac{n-s}{s(s-1)}\right)\hat{x}_{i,k} - \frac{1}{n}\sum_{j=1}^{n}\hat{x}_{j,k}$$

$$= \hat{x}_{i,k} - \frac{1}{n}\sum_{j=1}^{n}\hat{x}_{j,k},$$

as desired.

Now, we want to find $\omega$ such that equation 17 holds or, equivalently,

$$\mathbb{E}\left[\sum_{i=1}^{n}\left\|d_i^t\right\|^2\right] \le (1+\omega)\sum_{i=1}^{n}\left\|\hat{x}_i^t - \frac{1}{n}\sum_{j=1}^{n}\hat{x}_j^t\right\|^2.$$

We can reason on the coordinates separately, or all at once to ease the notations: we have

$$\mathbb{E}\left[\sum_{i=1}^{n}\left\|d_i^t\right\|^2\right] = p\frac{s}{n}\sum_{i=1}^{n}\mathbb{E}_{\Omega:i\in\Omega}\left\|a\hat{x}_i^t - \frac{a}{s}\sum_{j\in\Omega}\hat{x}_j^t\right\|^2.$$

For every $i \in [n]$,

$$\mathbb{E}_{\Omega:i\in\Omega}\left\|a\hat{x}_i^t - \frac{a}{s}\sum_{j\in\Omega}\hat{x}_j^t\right\|^2 = \mathbb{E}_{\Omega:i\in\Omega}\left\|\left(a - \frac{a}{s}\right)\hat{x}_i^t - \frac{a}{s}\sum_{j\in\Omega\setminus\{i\}}\hat{x}_j^t\right\|^2$$

$$= \left\|\left(a - \frac{a}{s}\right)\hat{x}_i^t\right\|^2 + \mathbb{E}_{\Omega:i\in\Omega}\left\|\frac{a}{s}\sum_{j\in\Omega\setminus\{i\}}\hat{x}_j^t\right\|^2$$

$$- 2\left\langle\left(a - \frac{a}{s}\right)\hat{x}_i^t, \frac{a}{s}\mathbb{E}_{\Omega:i\in\Omega}\sum_{j\in\Omega\setminus\{i\}}\hat{x}_j^t\right\rangle.$$

We have

$$\mathbb{E}_{\Omega:i\in\Omega}\sum_{j\in\Omega\setminus\{i\}}\hat{x}_j^t = \frac{s-1}{n-1}\sum_{j\in[n]\setminus\{i\}}\hat{x}_j^t = \frac{s-1}{n-1}\left(\sum_{j=1}^{n}\hat{x}_j^t - \hat{x}_i^t\right)$$

and

$$\mathbb{E}_{\Omega:i\in\Omega}\left\|\sum_{j\in\Omega\setminus\{i\}}\hat{x}_j^t\right\|^2 = \mathbb{E}_{\Omega:i\in\Omega}\sum_{j\in\Omega\setminus\{i\}}\left\|\hat{x}_j^t\right\|^2 + \mathbb{E}_{\Omega:i\in\Omega}\sum_{j\in\Omega\setminus\{i\}}\sum_{j'\in\Omega\setminus\{i,j\}}\left\langle\hat{x}_j^t, \hat{x}_{j'}^t\right\rangle$$

$$= \frac{s-1}{n-1}\sum_{j\in[n]\setminus\{i\}}\left\|\hat{x}_j^t\right\|^2 + \frac{s-1}{n-1}\frac{s-2}{n-2}\sum_{j\in[n]\setminus\{i\}}\sum_{j'\in[n]\setminus\{i,j\}}\left\langle\hat{x}_j^t, \hat{x}_{j'}^t\right\rangle$$

$$= \frac{s-1}{n-1}\left(1 - \frac{s-2}{n-2}\right)\sum_{j\in[n]\setminus\{i\}}\left\|\hat{x}_j^t\right\|^2 + \frac{s-1}{n-1}\frac{s-2}{n-2}\left\|\sum_{j\in[n]\setminus\{i\}}\hat{x}_j^t\right\|^2$$

$$= \frac{s-1}{n-1}\frac{n-s}{n-2}\left(\sum_{j=1}^{n}\left\|\hat{x}_j^t\right\|^2 - \left\|\hat{x}_i^t\right\|^2\right) + \frac{s-1}{n-1}\frac{s-2}{n-2}\left\|\sum_{j=1}^{n}\hat{x}_j^t - \hat{x}_i^t\right\|^2.$$

Hence,

$$
\begin{aligned}
\mathbb{E}\left[\sum_{i=1}^{n}\left\|d_i^t\right\|^2\right] = {} & p\frac{s}{n}\sum_{i=1}^{n}\left\|\left(a-\frac{a}{s}\right)\hat{x}_i^t\right\|^2 + ps\frac{a^2}{(s)^2}\frac{s-1}{n-1}\frac{n-s}{n-2}\sum_{j=1}^{n}\left\|\hat{x}_j^t\right\|^2 \\
& - p\frac{s}{n}\frac{a^2}{(s)^2}\frac{s-1}{n-1}\frac{n-s}{n-2}\sum_{i=1}^{n}\left\|\hat{x}_i^t\right\|^2 + p\frac{s}{n}\frac{a^2}{(s)^2}\frac{s-1}{n-1}\frac{s-2}{n-2}\sum_{i=1}^{n}\left\|\sum_{j=1}^{n}\hat{x}_j^t-\hat{x}_i^t\right\|^2 \\
& - 2p\frac{s}{n}\frac{a}{s}\frac{s-1}{n-1}\left(a-\frac{a}{s}\right)\sum_{i=1}^{n}\left\langle\hat{x}_i^t,\sum_{j=1}^{n}\hat{x}_j^t-\hat{x}_i^t\right\rangle \\
= {} & \frac{(n-1)^2}{psn}\sum_{i=1}^{n}\left\|\hat{x}_i^t\right\|^2 + \frac{(n-1)^2}{ps(s-1)n}\frac{n-s}{n-2}\sum_{i=1}^{n}\left\|\hat{x}_i^t\right\|^2 \\
& + \frac{1}{ps}\frac{s-2}{s-1}\frac{n-1}{n-2}\left\|\sum_{i=1}^{n}\hat{x}_i^t\right\|^2 - 2\frac{1}{psn}\frac{s-2}{s-1}\frac{n-1}{n-2}\left\|\sum_{i=1}^{n}\hat{x}_i^t\right\|^2 \\
& + \frac{1}{psn}\frac{s-2}{s-1}\frac{n-1}{n-2}\sum_{i=1}^{n}\left\|\hat{x}_i^t\right\|^2 + 2\frac{n-1}{psn}\sum_{i=1}^{n}\left\|\hat{x}_i^t\right\|^2 - 2\frac{n-1}{psn}\left\|\sum_{i=1}^{n}\hat{x}_i^t\right\|^2 \\
= {} & \frac{(n-1)(n+1)}{psn}\sum_{i=1}^{n}\left\|\hat{x}_i^t\right\|^2 + \frac{(n-1)^2}{ps(s-1)n}\frac{n-s}{n-2}\sum_{i=1}^{n}\left\|\hat{x}_i^t\right\|^2 \\
& - \frac{n-1}{psn}\frac{s}{s-1}\left\|\sum_{i=1}^{n}\hat{x}_i^t\right\|^2 + \frac{1}{psn}\frac{s-2}{s-1}\frac{n-1}{n-2}\sum_{i=1}^{n}\left\|\hat{x}_i^t\right\|^2 \\
= {} & \frac{(n^2-1)(s-1)(n-2)+(n-1)^2(n-s)+(s-2)(n-1)}{ps(s-1)n(n-2)}\sum_{i=1}^{n}\left\|\hat{x}_i^t\right\|^2 \\
& - \frac{n-1}{p(s-1)n}\left\|\sum_{i=1}^{n}\hat{x}_i^t\right\|^2 \\
= {} & \frac{n-1}{p(s-1)}\sum_{i=1}^{n}\left\|\hat{x}_i^t\right\|^2 - \frac{n-1}{p(s-1)n}\left\|\sum_{i=1}^{n}\hat{x}_i^t\right\|^2 \\
= {} & \frac{n-1}{p(s-1)}\sum_{i=1}^{n}\left\|\hat{x}_i^t - \frac{1}{n}\sum_{j=1}^{n}\hat{x}_j^t\right\|^2 .
\end{aligned}
$$

Therefore, we can set

$$
\omega := \frac{n-1}{p(s-1)} - 1. \tag{26}
$$

## D    PROOF OF THEOREM 3

We suppose that the assumptions in Theorem 3 hold. $s$ is set as the maximum of three values. Let us consider these three cases.

1) Suppose that $s = 2$. Since $2 = s \geq \lfloor cn \rfloor$ and $2 = s \geq \lfloor \frac{n}{d} \rfloor$, we have $c \leq \frac{3}{n}$ and $1 \leq \frac{3d}{n}$. Hence,

$$\mathcal{O}\left(\sqrt{\frac{n\kappa}{s}} + \frac{n}{s}\right)\left(\frac{sd}{n} + 1 + cd\right)$$
$$= \mathcal{O}\left(\sqrt{n\kappa} + n\right)\left(\frac{d}{n} + \frac{d}{n} + \frac{d}{n}\right)$$
$$= \mathcal{O}\left(\frac{d\sqrt{\kappa}}{\sqrt{n}} + d\right). \tag{27}$$

2) Suppose that $s = \lfloor \frac{n}{d} \rfloor$. Then $\frac{sd}{n} \leq 1$. Since $s \geq \lfloor cn \rfloor$ and $\lfloor \frac{n}{d} \rfloor = s \geq 2$, we have $cn \leq s + 1 \leq \frac{n}{d} + 1$ and $\frac{d}{n} \leq \frac{1}{2}$, so that $cd \leq 1 + \frac{d}{n} \leq 2$. Hence,

$$\mathcal{O}\left(\sqrt{\frac{n\kappa}{s}} + \frac{n}{s}\right)\left(\frac{sd}{n} + 1 + cd\right)$$
$$= \mathcal{O}\left(\sqrt{\frac{n\kappa}{s}} + \frac{n}{s}\right).$$

Since $2s \geq \frac{n}{d}$, we have $\frac{1}{s} \leq \frac{2d}{n}$ and

$$\mathcal{O}\left(\sqrt{\frac{n\kappa}{s}} + \frac{n}{s}\right)\left(\frac{sd}{n} + 1 + cd\right)$$
$$= \mathcal{O}\left(\sqrt{d\kappa} + d\right). \tag{28}$$

3) Suppose that $s = \lfloor cn \rfloor$. Then $s \leq cn$. Also, $2s \geq cn$ and $\frac{1}{s} \leq \frac{2}{cn}$. Since $s = \lfloor cn \rfloor \geq \lfloor \frac{n}{d} \rfloor$, we have $cn + 1 \geq \frac{n}{d}$ and $1 \leq cd + \frac{d}{n}$. Since $s = \lfloor cn \rfloor \geq 2$, we have $\frac{1}{n} \leq \frac{c}{2}$ and $1 \leq 2cd$. Hence,

$$\mathcal{O}\left(\sqrt{\frac{n\kappa}{s}} + \frac{n}{s}\right)\left(\frac{sd}{n} + 1 + cd\right)$$
$$= \mathcal{O}\left(\sqrt{\frac{\kappa}{c}} + \frac{1}{c}\right)(cd + cd + cd)$$
$$= \mathcal{O}\left(\sqrt{c\kappa}d + d\right). \tag{29}$$

By adding up the three upper bounds equation 27, equation 28, equation 29, we obtain the upper bound in equation 14.

## E    PROOF OF THEOREM 2

We suppose that the assumptions in Theorem 2 hold. A solution $x^\star \in \mathbb{R}^d$ to equation 1, which is supposed to exist, satisfies $\nabla f(x^\star) = \frac{1}{n}\sum_{i=1}^n \nabla f_i(x^\star) = 0$. $x^\star$ is not necessarily unique but $h_i^\star := \nabla f_i(x^\star)$ is unique.

We define the Bregman divergence of a $L$-smooth convex function $g$ at points $x, x' \in \mathbb{R}^d$ as $D_g(x, x') := g(x) - g(x') - \langle \nabla g(x'), x - x' \rangle \geq 0$. We have $D_g(x, x') \geq \frac{1}{2L}\|\nabla g(x) - \nabla g(x')\|^2$. We can note that for every $x \in \mathbb{R}^d$ and $i = 1, \ldots, n$, $D_{f_i}(x, x^\star)$ is the same whatever the solution $x^\star$.

For every $t \geq 0$, we define the Lyapunov function

$$\Psi^t := \frac{1}{\gamma}\sum_{i=1}^n \|x_i^t - x^\star\|^2 + \frac{\gamma}{p^2\eta}\frac{n-1}{s-1}\sum_{i=1}^n \|h_i^t - h_i^\star\|^2, \tag{30}$$

Starting from equation 22 with the substitutions detailed at the end of the proof of Theorem 1, we have, for every $t \geq 0$,

$$\mathbb{E}\big[\Psi^{t+1} \mid \mathcal{F}_t\big] = \frac{1}{\gamma} \sum_{i=1}^{n} \mathbb{E}\Big[\big\|x_i^{t+1} - x^\star\big\|^2 \mid \mathcal{F}_t\Big] + \frac{\gamma}{p^2\eta} \frac{n-1}{s-1} \sum_{i=1}^{n} \mathbb{E}\Big[\big\|h_i^{t+1} - h_i^\star\big\|^2 \mid \mathcal{F}_t\Big]$$

$$\leq \frac{1}{\gamma} \sum_{i=1}^{n} \big\|\big(x_i^t - \gamma\nabla f_i(x_i^t)\big) - \big(x^\star - \gamma\nabla f_i(x^\star)\big)\big\|^2$$

$$+ \left(\frac{\gamma}{p^2\eta} \frac{n-1}{s-1} - \gamma\right) \sum_{i=1}^{n} \big\|h_i^t - h_i^\star\big\|^2 + \frac{p}{\gamma}(\eta - 1 + \nu) \sum_{i=1}^{n} \left\|\hat{x}_i^t - \frac{1}{n}\sum_{j=1}^{n} \hat{x}_j^t\right\|^2$$

with

$$\big\|\big(x_i^t - \gamma\nabla f_i(x_i^t)\big) - \big(x^\star - \gamma\nabla f_i(x^\star)\big)\big\|^2 = \big\|x_i^t - x^\star\big\|^2 - 2\gamma\langle\nabla f_i(x_i^t) - \nabla f_i(x^\star), x_i^t - x^\star\rangle$$

$$+ \gamma^2 \big\|\nabla f_i(x_i^t) - \nabla f_i(x^\star)\big\|^2$$

$$\leq \big\|x_i^t - x^\star\big\|^2 - (2\gamma - \gamma^2 L)\langle\nabla f_i(x_i^t) - \nabla f_i(x^\star), x_i^t - x^\star\rangle,$$

where the second inequality follows from cocoercivity of the gradient. Moreover, for every $x, x'$, $D_{f_i}(x, x') \leq \langle\nabla f_i(x) - \nabla f_i(x'), x - x'\rangle$. Therefore,

$$\mathbb{E}\big[\Psi^{t+1} \mid \mathcal{F}_t\big] \leq \Psi^t - (2 - \gamma L) \sum_{i=1}^{n} D_{f_i}(x_i^t, x^\star)$$

$$- \gamma \sum_{i=1}^{n} \big\|h_i^t - h_i^\star\big\|^2 + \frac{p}{\gamma}(\eta - 1 + \nu) \sum_{i=1}^{n} \left\|\hat{x}_i^t - \frac{1}{n}\sum_{j=1}^{n} \hat{x}_j^t\right\|^2 .$$

Telescopic the sum and using the tower rule of expectations, we get summability over $t$ of the three negative terms above: for every $T \geq 0$, we have

$$(2 - \gamma L) \sum_{i=1}^{n} \sum_{t=0}^{T} \mathbb{E}\big[D_{f_i}(x_i^t, x^\star)\big] \leq \Psi^0 - \mathbb{E}\big[\Psi^{T+1}\big] \leq \Psi^0, \tag{31}$$

$$\gamma \sum_{i=1}^{n} \sum_{t=0}^{T} \mathbb{E}\Big[\big\|h_i^t - h_i^\star\big\|^2\Big] \leq \Psi^0 - \mathbb{E}\big[\Psi^{T+1}\big] \leq \Psi^0, \tag{32}$$

$$\frac{p}{\gamma}(1 - \nu - \eta) \sum_{i=1}^{n} \sum_{t=0}^{T} \mathbb{E}\left[\left\|\hat{x}_i^t - \frac{1}{n}\sum_{j=1}^{n} \hat{x}_j^t\right\|^2\right] \leq \Psi^0 - \mathbb{E}\big[\Psi^{T+1}\big] \leq \Psi^0. \tag{33}$$

Taking ergodic averages and using convexity of the squared norm and of the Bregman divergence, we can now get $\mathcal{O}(1/T)$ rates. We use a tilde to denote averages over the iterations so far. That is, for every $i = 1, \ldots, n$ and $T \geq 0$, we define

$$\tilde{x}_i^T := \frac{1}{T+1} \sum_{t=0}^{T} x_i^t$$

and

$$\tilde{x}^T := \frac{1}{n} \sum_{i=1}^{n} \tilde{x}_i^T.$$

The Bregman divergence is convex in its first argument, so that, for every $T \geq 0$,

$$\sum_{i=1}^{n} D_{f_i}(\tilde{x}_i^T, x^\star) \leq \sum_{i=1}^{n} \frac{1}{T+1} \sum_{t=0}^{T} D_{f_i}(x_i^t, x^\star).$$

Combining this inequality with equation 31 yields, for every $T \geq 0$,

$$(2 - \gamma L) \sum_{i=1}^{n} \mathbb{E}\left[D_{f_i}(\tilde{x}_i^T, x^\star)\right] \leq \frac{\Psi^0}{T+1}. \tag{34}$$

Similarly, for every $i = 1, \ldots, n$ and $T \geq 0$, we define

$$\tilde{h}_i^T := \frac{1}{T+1} \sum_{t=0}^{T} h_i^t$$

and we have, for every $T \geq 0$,

$$\sum_{i=1}^{n} \left\| \tilde{h}_i^T - h_i^\star \right\|^2 \leq \sum_{i=1}^{n} \frac{1}{T+1} \sum_{t=0}^{T} \left\| h_i^t - h_i^\star \right\|^2.$$

Combining this inequality with equation 32 yields, for every $T \geq 0$,

$$\gamma \sum_{i=1}^{n} \mathbb{E}\left[\left\| \tilde{h}_i^T - h_i^\star \right\|^2\right] \leq \frac{\Psi^0}{T+1}. \tag{35}$$

Finally, for every $i = 1, \ldots, n$ and $T \geq 0$, we define

$$\tilde{\hat{x}}_i^T := \frac{1}{T+1} \sum_{t=0}^{T} \hat{x}_i^t$$

and

$$\tilde{\hat{x}}^T := \frac{1}{n} \sum_{i=1}^{n} \tilde{\hat{x}}_i^T,$$

and we have, for every $T \geq 0$,

$$\sum_{i=1}^{n} \left\| \tilde{\hat{x}}_i^T - \tilde{\hat{x}}^T \right\|^2 \leq \sum_{i=1}^{n} \frac{1}{T+1} \sum_{t=0}^{T} \left\| \hat{x}_i^t - \frac{1}{n} \sum_{j=1}^{n} \hat{x}_j^t \right\|^2.$$

Combining this inequality with equation 33 yields, for every $T \geq 0$,

$$\frac{p}{\gamma}(1 - \nu - \eta) \sum_{i=1}^{n} \mathbb{E}\left[\left\| \tilde{\hat{x}}_i^T - \tilde{\hat{x}}^T \right\|^2\right] \leq \frac{\Psi^0}{T+1}. \tag{36}$$

Next, we have, for every $i = 1, \ldots, n$ and $T \geq 0$,

$$\left\| \nabla f(\tilde{x}_i^T) \right\|^2 \leq 2 \left\| \nabla f(\tilde{x}_i^T) - \nabla f(\tilde{x}^T) \right\|^2 + 2 \left\| \nabla f(\tilde{x}^T) \right\|^2$$
$$\leq 2L^2 \left\| \tilde{x}_i^T - \tilde{x}^T \right\|^2 + 2 \left\| \nabla f(\tilde{x}^T) \right\|^2. \tag{37}$$

Moreover, for every $T \geq 0$ and solution $x^\star$ to equation 1,

$$\left\| \nabla f(\tilde{x}^T) \right\|^2 = \left\| \nabla f(\tilde{x}^T) - \nabla f(x^\star) \right\|^2$$
$$\leq \frac{1}{n} \sum_{i=1}^{n} \left\| \nabla f_i(\tilde{x}^T) - \nabla f_i(x^\star) \right\|^2$$
$$\leq \frac{2}{n} \sum_{i=1}^{n} \left\| \nabla f_i(\tilde{x}^T) - \nabla f_i(\tilde{x}_i^T) \right\|^2 + \frac{2}{n} \sum_{i=1}^{n} \left\| \nabla f_i(\tilde{x}_i^T) - \nabla f_i(x^\star) \right\|^2$$
$$\leq \frac{2L^2}{n} \sum_{i=1}^{n} \left\| \tilde{x}_i^T - \tilde{x}^T \right\|^2 + \frac{4L}{n} \sum_{i=1}^{n} D_{f_i}(\tilde{x}_i^T, x^\star). \tag{38}$$

There remains to control the terms $\left\| \tilde{x}_i^T - \tilde{x}^T \right\|^2$: we have, for every $T \geq 0$,

$$\sum_{i=1}^n \left\| \tilde{x}_i^T - \tilde{x}^T \right\|^2 \leq 2 \sum_{i=1}^n \left\| (\tilde{x}_i^T - \tilde{x}^T) - (\hat{\tilde{x}}_i^T - \hat{\tilde{x}}^T) \right\|^2 + 2 \sum_{i=1}^n \left\| \hat{\tilde{x}}_i^T - \hat{\tilde{x}}^T \right\|^2$$

$$\leq 2 \sum_{i=1}^n \left\| \tilde{x}_i^T - \hat{\tilde{x}}_i^T \right\|^2 + 2 \sum_{i=1}^n \left\| \hat{\tilde{x}}_i^T - \hat{\tilde{x}}^T \right\|^2. \tag{39}$$

For every $i = 1, \ldots, n$ and $t \geq 0$,

$$\hat{x}_i^t = x_i^t - \gamma \left( \nabla f_i(x_i^t) - h_i^t \right)$$

so that, for every $i = 1, \ldots, n$ and $T \geq 0$,

$$\tilde{x}_i^T - \hat{\tilde{x}}_i^T = \gamma \frac{1}{T+1} \sum_{t=0}^T \nabla f_i(x_i^t) - \gamma \tilde{h}_i^T$$

and

$$\left\| \tilde{x}_i^T - \hat{\tilde{x}}_i^T \right\|^2 = \gamma^2 \left\| \frac{1}{T+1} \sum_{t=0}^T \nabla f_i(x_i^t) - \tilde{h}_i^T \right\|^2$$

$$\leq 2\gamma^2 \frac{1}{T+1} \sum_{t=0}^T \left\| \nabla f_i(x_i^t) - \nabla f_i(x^\star) \right\|^2 + 2\gamma^2 \left\| \tilde{h}_i^T - h_i^\star \right\|^2$$

$$\leq 4L\gamma^2 \frac{1}{T+1} \sum_{t=0}^T D_{f_i}(x_i^t, x^\star) + 2\gamma^2 \left\| \tilde{h}_i^T - h_i^\star \right\|^2. \tag{40}$$

Combining equation 37, equation 38, equation 39, equation 40, we get, for every $T \geq 0$,

$$\sum_{i=1}^n \left\| \nabla f(\tilde{x}_i^T) \right\|^2 \leq 2L^2 \sum_{i=1}^n \left\| \tilde{x}_i^T - \tilde{x}^T \right\|^2 + 2n \left\| \nabla f(\tilde{x}^T) \right\|^2$$

$$\leq 2L^2 \sum_{i=1}^n \left\| \tilde{x}_i^T - \tilde{x}^T \right\|^2 + 2L^2 \sum_{i=1}^n \left\| \tilde{x}_i^T - \tilde{x}^T \right\|^2 + 4L \sum_{i=1}^n D_{f_i}(\tilde{x}_i^T, x^\star)$$

$$= 4L^2 \sum_{i=1}^n \left\| \tilde{x}_i^T - \tilde{x}^T \right\|^2 + 4L \sum_{i=1}^n D_{f_i}(\tilde{x}_i^T, x^\star)$$

$$\leq 8L^2 \sum_{i=1}^n \left\| \tilde{x}_i^T - \hat{\tilde{x}}_i^T \right\|^2 + 8L^2 \sum_{i=1}^n \left\| \hat{\tilde{x}}_i^T - \hat{\tilde{x}}^T \right\|^2 + 4L \sum_{i=1}^n D_{f_i}(\tilde{x}_i^T, x^\star)$$

$$\leq 32L^3 \gamma^2 \frac{1}{T+1} \sum_{i=1}^n \sum_{t=0}^T D_{f_i}(x_i^t, x^\star) + 16L^2 \gamma^2 \sum_{i=1}^n \left\| \tilde{h}_i^T - h_i^\star \right\|^2$$

$$+ 8L^2 \sum_{i=1}^n \left\| \hat{\tilde{x}}_i^T - \hat{\tilde{x}}^T \right\|^2 + 4L \sum_{i=1}^n D_{f_i}(\tilde{x}_i^T, x^\star).$$

Taking the expectation and using equation 31, equation 35, equation 36 and equation 34, we get, for every $T \geq 0$,

$$
\begin{aligned}
\sum_{i=1}^{n} \mathbb{E}\left[\left\|\nabla f(\tilde{x}_i^T)\right\|^2\right] \leq{} & 32L^3\gamma^2 \frac{1}{T+1} \sum_{i=1}^{n} \sum_{t=0}^{T} \mathbb{E}\left[D_{f_i}(x_i^t, x^\star)\right] \\
& + 16L^2\gamma^2 \sum_{i=1}^{n} \mathbb{E}\left[\left\|\tilde{h}_i^T - h_i^\star\right\|^2\right] \\
& + 8L^2 \sum_{i=1}^{n} \mathbb{E}\left[\left\|\hat{\tilde{x}}_i^T - \tilde{x}^T\right\|^2\right] + 4L \sum_{i=1}^{n} \mathbb{E}\left[D_{f_i}(\tilde{x}_i^T, x^\star)\right]. \\
\leq{} & \frac{32L^3\gamma^2}{2-\gamma L}\frac{\Psi^0}{T+1} + 16L^2\gamma\frac{\Psi^0}{T+1} + \frac{8L^2\gamma}{p(1-\nu-\eta)}\frac{\Psi^0}{T+1} + \frac{4L}{2-\gamma L}\frac{\Psi^0}{T+1} \\
={} & \left[\frac{32L^3\gamma^2 + 4L}{2-\gamma L} + 16L^2\gamma + \frac{8L^2\gamma}{p(1-\nu-\eta)}\right]\frac{\Psi^0}{T+1}.
\end{aligned}
$$

Hence, with $\gamma = \Theta\left(\frac{p}{L}\right)$ and a fixed $\eta$, we have

$$
\sum_{i=1}^{n} \mathbb{E}\left[\left\|\nabla f(\tilde{x}_i^T)\right\|^2\right] \leq \epsilon
$$

after

$$
\mathcal{O}\left(\frac{\Psi^0}{p\epsilon}\right)
$$

iterations and

$$
\mathcal{O}\left(\frac{\Psi^0}{\epsilon}\right)
$$

communication rounds.

We note that in these conditions, with $\gamma$ scaled by $p$, LT does not yield any acceleration: the communication complexity is the same whatever $p$. CC is effective, however, since we communicate much less than $d$ floats during every communication round.

