# OpenReview forum: "Provably Doubly Accelerated Federated Learning: The First Theoretically Successful Combination of Local Training and Communication Compression"
_ICLR.cc/2024/Conference — ICLR 2024 Conference Desk Rejected Submission_

### Official Review · Reviewer_Va6u · 2023-10-30

**Soundness:** 3 good
**Presentation:** 2 fair
**Contribution:** 3 good
**Rating:** 6
**Confidence:** 3

**Summary:**

The paper studied how to accelerate a federated optimization method using local training and communication compression.

To that end, the author proposed CompressedScaffnew and proved that this algorithm achieves the total communication complexity of $\widetilde{O}(\sqrt{d \kappa} + d)$ for the strongly convex case, which is the current SOTA result.

**Strengths:**

The paper is mostly easy to follow, while some parts are vague and hard to understand.

The theoretical result in the paper is new and exciting. It is good to see that acceleration, message compression, and local training could be combined organically.

The numerical experiments support the theories.

**Weaknesses:**

1. No lower bound for the communication complexity.

2. Some parts are not very clear (see Question for more details).

3. No conclusion in the paper. Possible extensions are not discussed.

4. Discussion on generally convex cases seems not to be sufficient as the strongly convex case. The empirical experiments are only for the strongly convex case.

**Questions:**

1. At the beginning of the second paragraph of Section 2, the author claims that ``It is very challenging to combine LT and CC.``
However, the proposed algorithm is somewhat quite similar to Scaffnew. It makes me wonder what the difficulty is, while the solution is not a huge deviation from a previous algorithm. I found that the author tried to explain this point at the end of Page 4. However, I found this part really hard to grasp, given that the algorithm CompressedScaffnew is introduced in the next section, and many terminologies are not formally defined (such as ``control variates`` and ``two mechanisms``). I hope the author could elaborate more on the difficulties, which would make readers more clear on the novelty.

2. Is there any insight for the choice of $\eta$ in (7)? Why is it in this specific form?

3. I know the main focus of this paper is on convex deterministic optimization. However, I would like to whether the proposed technique could be extended to stochastic /non-convex optimization. This would definitely benefit future work.

4. Most of the focus was put on the strongly convex case. Note that the author also provides the convergence rate for the generally convex case. However, the author didn't discuss the communication complexity of that case.  What is the corresponding communication complexity? Is it still better than the baseline methods?

5. Another question that puzzles me is that in Theorem 2, the author studied the convergence of gradient norm $\|\nabla f(\tilde{x}_i^T)\|$
 rather than the objective loss $f(\tilde{x}_i^T) - f(x^{\star})$. Is there any difficulty to do so?

I will increase my point if the author could address my questions well.

---

> ### Author Response · Authors · 2023-11-16
> **Response part 1/2**
>
> Thank you for your thorough analysis and positive evaluation of our work.
>
> > No lower bound for the communication complexity.
>
> we are not aware of existing works establishing lower bounds for our setting, and we conjecture that the optimal communication complexity in number of reals is $\mathcal{O}(d\sqrt{\kappa}/\sqrt{n}+\sqrt{d}\sqrt{\kappa}+d)$, as shown in green in Table 1 for $c=0$. Papers such as "The Min-Max Complexity of Distributed Stochastic Convex Optimization with Intermittent Communication", 2021, or "Information-theoretic lower bounds on the Oracle complexity of convex optimization", 2012, provide lower bounds for other problem classes. Thus, we conjecture that CompressedScaffnew is optimal for $c=0$. For $c>0$, it is certainly not optimal since there is no downlink compression. We leave the even more challenging topic of combining LT with bidirectional compression for future work.
>
> > No conclusion in the paper. Possible extensions are not discussed.
>
> Yes, by lack of space we had put the description of the experiments in Appendix A and omitted the conclusion.
> We have revised the paper and added a conclusion with venues for future work, in Appendix B.
>
> > Discussion on generally convex cases seems not to be sufficient as the strongly convex case.
>
> We have added a paragraph at the end of Appendix E. In the general convex regime, we can still demonstrate acceleration from compression, but not from local training.
>
> > At the beginning of the second paragraph of Section 2, the author claims that "It is very challenging to combine LT and CC".
>
> Yes, given that FedAvg has been presented in 2016 and studied in thousands of papers, and the modern compression algorithm DIANA has been proposed in 2019 and has been studied extensively too, we believe that if this topic was not challenging, this combination would have been done before.
>
>
> > However, the proposed algorithm is somewhat quite similar to Scaffnew / the solution is not a huge deviation from a previous algorithm / It makes me wonder what the difficulty is, while  I found that the author tried to explain this point at the end of Page 4.
>
> CompressedScaffnew reverts to Scaffnew without compression, but our contribution is precisely compression, and the compression mechanism we propose is very specific and far from trivial. Another key element is the update of the control variates $h_i$: only their coordinates which have been communicated and contribute to the new model estimate $\bar{x}^t$ are updated. Another important property is that $\sum_i h_i^t$ always remains zero. We explain these aspects in Section 2. We can also mention that the proof of Theorem 1 in Appendix C is 6 page long. Plugging compression into Scaffnew in any other way does not work, different ideas have been tried and the obtained algorithms diverge numerically.
>
>
> > many terminologies are not formally defined (such as "control variates" and "two mechanisms").
>
> "two mechanisms" is not a technical term, we are referring to the 2 mechanisms to update the model $x$ and the control variates $h_i$, respectively. "control variates" is a common term in the literature of variance-reduced algorithms, such as SAGA, DIANA, error feedback, gradient tracking... These variables offset the random noise,  so that the algorithms converge to the exact solution, and not to a neighborhood like with non-variance-reduced SGD-type algorithms.
>
> > Is there any insight for the choice of in (7)? Why is it in this specific form?
>
> This form is explained in the paragraph before Section C.1. in Appendix C. It comes from the variance due to sampling among the $n$ clients, shown in (21).
>
> > I know the main focus of this paper is on convex deterministic optimization. However, I would like to whether the proposed technique could be extended to stochastic /non-convex optimization. This would definitely benefit future work.
>
> Extending the algorithm to stochastic instead of exact gradients is a venue for future work, as written in the conclusion. This should not be very difficult, but we have to study how the noise of the stochastic gradients propagates when applying the further random mechanism of compression. This deserves a thorough investigation. Regarding the nonconvex setting, this requires different proof techniques, typically based on the decay of the objective function, and we currently don't know how to do it. To the best of our knowledge, Scaffnew has not been analyzed for nonconvex problems, this is probably very difficult.

---

> ### Author Response · Authors · 2023-11-16
> **Response part 2/2**
>
> > the author also provides the convergence rate for the generally convex case. However, the author didn't discuss the communication complexity of that case. What is the corresponding communication complexity? Is it still better than the baseline methods?
>
> As replied above, we have added a paragraph at the end of Appendix E. The rate is $\mathcal{O}(1/t)$, which is not accelerated (accelerated gradient descent has rate $\mathcal{O}(1/t^2)$). However, the dependence with respect to the dimension $d$ is reduced, thanks to compression. Contrary to the analysis in the strongly convex setting, which is certainly tight, we believe that our analysis in the general convex case is loose, that is why we do not put much emphasis on it. Further insights will probably come from a future analysis in the nonconvex setting.
>
> > in Theorem 2, the author studied the convergence of gradient norm rather than the objective loss. Is there any difficulty to do so?
>
> Yes, this is difficult. If we were able to study the decay of the objective loss, we would be able to tackle the nonconvex setting, because the descent lemma for smooth functions is all you need, and convexity is not needed in such proof techniques. But the difficulty comes from handling the update of the control variates. More generally, we can interpret the control variates as dual variables, a perspective studied in Condat and Richtárik, "RandProx: Primal-Dual Optimization Algorithms with Randomized Proximal Updates”, ICLR 2023. It is notoriously difficult to prove convergence of primal-dual algorithms in nonconvex settings.

---

### Official Review · Reviewer_5itr · 2023-11-01

**Soundness:** 3 good
**Presentation:** 4 excellent
**Contribution:** 3 good
**Rating:** 6
**Confidence:** 3

**Summary:**

The paper presents a new algorithm for FL with communication compression and local steps. Authors provide theoretical analysis that shows linear convergence and outperform other SOTA approaches. Double acceleration $\tilde{O}(\sqrt{\kappa}\sqrt{d})$ is achieved by local training and specific compressor. Simple experiments on logistics with theoretical hyperparameters show the superiority of proposed method compared to Scaffnew and GD.

**Strengths:**

1. Strong theoretical result, that improves complexities of SOTA methods with no assumptions on similarity, relying solely on the assumption of strong convexity and smoothness.
2. Double acceleration is theoretically achieved by combination of local steps and compression.
3. Paper is well written and efficiently presents mathematical details.

**Weaknesses:**

1. Experiments. The comparison is solely between GD, Scaffnew, and the proposed method (Scaffnew + CC). It is noteworthy that the absence of a comparison with methods incorporating communication compression ([1, 2]) and other SOTA approaches limits the comprehensiveness of the experimental evaluation. Furthermore, there is no surprise that the proposed method outperforms Scaffnew as the authors claim (page 8) that without compression the proposed algorithm achieves Scaffnew rate. Additional experiments with other baselines are warranted to provide a more comprehensive analysis
2. Experiments. It would be beneficial for the authors to conduct an additional set of experiments with fine-tuned hyperparameters (HP), as theoretical HP are typically unknown during the optimization process
3. Compression. The proposed framework works with specific compressor and does not allow for famous and practically successful compression techniques like Top-K and random dithering.

[1] Horváth, Samuel, et al. "Stochastic distributed learning with gradient quantization and double-variance reduction." Optimization Methods and Software 38.1 (2023): 91-106.

[2] Haddadpour, Farzin, et al. "Federated learning with compression: Unified analysis and sharp guarantees." International Conference on Artificial Intelligence and Statistics. PMLR, 2021.

**Questions:**

1. Title. Why the paper is called "The First Theoretically Successful Combination of Local Training and Communication Compression" if there already exist combination of local steps and communication compression with theoretical guarantees, e.g. [1, 2]?
2. Experiments. I wonder how the convergence changes with different choice of $p$, as it is allows to balance between local training and communication.



[1] Horváth, Samuel, et al. "Stochastic distributed learning with gradient quantization and double-variance reduction." Optimization Methods and Software 38.1 (2023): 91-106.

[2] Haddadpour, Farzin, et al. "Federated learning with compression: Unified analysis and sharp guarantees." International Conference on Artificial Intelligence and Statistics. PMLR, 2021.

---

> ### Author Response · Authors · 2023-11-16
> **Response**
>
> Thank you for your positive evaluation of our work and for acknowledging the significance of our contributions. So we are surprised of your score "2 fair".
>
> > the absence of a comparison with methods incorporating communication compression ([1, 2]) and other SOTA approaches limits the comprehensiveness of the experimental evaluation
>
> The algorithm in [2] is Scaffold, a popular algorithm. The accelerated algorithm Scaffnew of Mishchenko et al., presented at ICML 2022, improves significantly upon Scaffold, as the play on words in its name suggests. The experiments in their paper show that Scaffnew outperforms Scaffold in all regimes, as the theory predicts, since Scaffold is not accelerated. So, Scaffnew can be considered the current state of the art of algorithms communicating full vectors, so without compression. That is why we used Scaffnew as the baseline of LT but no CC in our experiments.
>
> On the other hand, the algorithm DIANA, developed by the same team of Mishchenko et al. in 2019 and generalized in several papers including [1], features CC but no LT. We describe its properties in our paper, and as summarized in footnote (a) of Table 1, CompressedScaffnew has a better theoretical complexity than DIANA. It is important to note that DIANA has uplink compression but communicates full-dimensional vectors at every iteration. So, unless $c=0$ exactly, which means that downlink communication is completely ignored, even with a small $c$, the total communication complexity of DIANA is huge, as it is dominated by non-compressed downlink communication. So, the curves for DIANA in our experiments would be essentially $1/c$ times better than the ones of GD, i.e. 5 times better in Figures 2 and 3, which is negligible. This is reflected by the factor $c d^2$ in the complexity of DIANA in Table 1. That is why we did not include DIANA in our experiments.
>
> >  It would be beneficial for the authors to conduct an additional set of experiments with fine-tuned hyperparameters (HP)
>
> We are not able to run such experiments in the limited timeframe of this rebuttal, but upon acceptance, we will do it and add the results in the final version of the paper.
>
> > The proposed framework works with specific compressor and does not allow for famous and practically successful compression techniques like Top-K and random dithering.
>
> This is correct, as of now the method only works with the specific compressor we designed for this purpose. It is linear and unbiased, which is necessary for our derivations. We do not know how to use any other compressor. Of course, we may always wish we could do better, and there is certainly room for improvement. On the other hand, the proposed compressor is enough to achieve the doubly accelerated complexity, shown in green in Table 1, which is a premiere, and which we conjecture is optimal if $c=0$.
>
> > Why the paper is called "The First Theoretically Successful Combination of Local Training and Communication Compression" if there already exist combination of local steps and communication compression with theoretical guarantees, e.g. [1, 2]?
>
> Scaffold in [1] has no compression, and DIANA in [2] has no local steps. As written in Section 2, "The only linearly converging LT + CC algorithm we are aware of is FedCOMGATE (Haddadpour et al., 2021). But its rate is $\mathcal{O}(d\kappa\log \epsilon^{-1})$, which does not show any acceleration." So, as the title says, CompressedScaffnew is the first algorithm with doubly accelerated complexity. That is why we stress that is is the first successful combination of the 2 techniques, since it retains their strengths, namely their two different acceleration effects, for the first time.
>
> > I wonder how the convergence changes with different choice of p, as it is allows to balance between local training and communication.
>
> There is indeed a tradeoff in the choice of $p$: we want to minimize $\sum_i f_i(x_i)$, but also enforce the consensus  $x_1=\cdots=x_n$. With a large $p$, the number of local steps is too small to obtain the acceleration from $\kappa$ to $\sqrt{\kappa}$. With a too small $p$, the extra local steps are harmful, because the local variables are attracted too much to the local minimizers $\arg \min f_i$ and deviate too much from each other (they are the same at the beginning of each round). This makes the update of the control variates of worse quality and slows down the convergence as a whole. In fact, in Scaffnew the communication complexity is $\max(p\kappa,1/p)$, that is why $p=1/\sqrt{\kappa}$ is the optimal choice.

---

> > ### Comment · Reviewer_5itr · 2023-11-20
> >
> > Thank you for clarification. It would be nice to see experiments with different $p$ and tuned HP.
> >
> > I changed my score accordingly.